# BEYOND LIPSCHITZ: SHARP GENERALIZATION AND EXCESS RISK BOUNDS FOR FULL-BATCH GD

**Konstantinos E. Nikolakakis**[*], **Farzin Haddadpour, Amin Karbasi**[1] **& Dionysios S. Kalogerias**
Department of Electrical Engineering
Yale University, Yale University & Google Research[1]
`{first.last}@yale.edu`

## ABSTRACT

We provide sharp path-dependent generalization and excess risk guarantees for the full-batch Gradient Descent (GD) algorithm on smooth losses (possibly non-Lipschitz, possibly nonconvex). At the heart of our analysis is an upper bound on the generalization error, which implies that *average output stability* and a *bounded expected optimization error at termination* lead to generalization. This result shows that a small generalization error occurs along the optimization path, and allows us to bypass Lipschitz or sub-Gaussian assumptions on the loss prevalent in previous works. For nonconvex, convex, and strongly convex losses, we show the explicit dependence of the generalization error in terms of the accumulated path-dependent optimization error, terminal optimization error, number of samples, and number of iterations. For nonconvex smooth losses, we prove that full-batch GD efficiently generalizes close to any stationary point at termination, and recovers the generalization error guarantees of stochastic algorithms with fewer assumptions. For smooth convex losses, we show that the generalization error is tighter than existing bounds for SGD (up to one order of error magnitude). Consequently the excess risk matches that of SGD for quadratically less iterations. Lastly, for strongly convex smooth losses, we show that full-batch GD achieves essentially the same excess risk rate as compared with the state of the art on SGD, but with an exponentially smaller number of iterations (logarithmic in the dataset size).

## 1 INTRODUCTION

Gradient based learning (Lecun et al., 1998) is a well established topic with a large body of literature on algorithmic generalization and optimization errors. For general smooth convex losses, optimization error guarantees have long been well-known (Nesterov, 1998). Similarly, Absil et al. (2005) and Lee et al. (2016) have showed convergence of Gradient Descent (GD) to minimizers and local minima for smooth nonconvex functions. More recently, Chatterjee (2022), Liu et al. (2022) and Allen-Zhu et al. (2019) established global convergence of GD for deep neural networks under appropriate conditions.

Generalization error analysis of stochastic training algorithms has recently gained increased attention. Hardt et al. (2016) showed uniform stability final-iterate bounds for vanilla Stochastic Gradient Descent (SGD). More recent works have developed alternative generalization error bounds with probabilistic guarantees (Feldman & Vondrak, 2018; 2019; Madden et al., 2020; Klochkov & Zhivotovskiy, 2021) and data-dependent variants (Kuzborskij & Lampert, 2018), or under weaker assumptions such as strongly quasi-convex (Gower et al., 2019), non-smooth convex (Feldman, 2016; Bassily et al., 2020; Lei & Ying, 2020b; Lei et al., 2021a), and pairwise losses (Lei et al., 2021b; 2020). In the nonconvex case, Zhou et al. (2021b) provide bounds that involve the on-average variance of the stochastic gradients. Generalization performance of other algorithmic variants lately gain further attention, including SGD with early momentum (Ramezani-Kebrya et al., 2021), randomized coordinate descent (Wang et al., 2021c), look-ahead approaches (Zhou et al., 2021a), noise injection methods (Xing et al., 2021), and stochastic gradient Langevin dynamics (Pensia et al., 2018; Mou et al., 2018; Li et al., 2020; Negrea et al., 2019; Zhang et al., 2021; Farghly & Rebeschini, 2021; Wang et al., 2021a;b).

---

[*]Lead & corresponding author

| Excess Risk Upper Bounds: GD vs SGD | | | | |
|---|---|---|---|---|
| Algorithm | Iterations | Interpolation | Bound | $\beta$-Smooth Loss |
| **GD (this work)** $\eta_t = 1/2\beta$ | $T = \sqrt{n}$ | No | $\mathcal{O}\left(\dfrac{1}{\sqrt{n}}\right)$ | Convex |
| SGD $\eta_t = 1/\sqrt{T}$ (Lei & Ying, 2020b) | $T = n$ | No | $\mathcal{O}\left(\dfrac{1}{\sqrt{n}}\right)$ | Convex |
| **GD (this work)** $\eta_t = 1/2\beta$ | $T = n$ | Yes | $\mathcal{O}\left(\dfrac{1}{n}\right)$ | Convex |
| SGD $\eta_t = 1/2\beta$ (Lei & Ying, 2020b) | $T = n$ | Yes | $\mathcal{O}\left(\dfrac{1}{n}\right)$ | Convex |
| **GD (this work)** $\eta_t = 2/(\beta + \gamma)$ | $T = \Theta(\log n)$ | No | $\mathcal{O}\left(\dfrac{\sqrt{\log(n)}}{n}\right)$ | $\gamma$-Strongly Convex (Objective) |
| SGD $\eta_t = 2/(t + t_0)\gamma$ (Lei & Ying, 2020b) | $T = \Theta(n)$ | No | $\mathcal{O}\left(\dfrac{1}{n}\right)$ | $\gamma$-Strongly Convex (Objective) |

Table 1: Comparison of the excess risk bounds for the full-batch GD and SGD algorithms by Lei & Ying (2020b, Corollary 5 & Theorem 11). We denote by $n$ the number of samples, $T$ the total number of iterations, $\eta_t$ the step size at time $t$, and by $\epsilon_{\mathbf{c}} \triangleq \mathbb{E}[R_S(W_S^*)]$ the interpolation error.

Even though many previous works consider stochastic training algorithms and some even suggest that stochasticity may be necessary (Hardt et al., 2016; Charles & Papailiopoulos, 2018) for good generalization, recent empirical studies have demonstrated that deterministic algorithms can indeed generalize well; see, e.g., (Hoffer et al., 2017; Geiping et al., 2022). In fact, Hoffer et al. (2017) showed empirically that for large enough number of iterations full-batch GD generalizes comparably to SGD. Similarly, Geiping et al. (2022) experimentally showed that strong generalization behavior is still observed in the absence of stochastic sampling. Such interesting empirical evidence reasonably raise the following question: "Are there problem classes for which deterministic training generalizes more efficiently than stochastic training?"

While prior works provide extensive analysis of the generalization error and excess risk of stochastic gradient methods, tight and path-dependent generalization error and excess risk guarantees in non-stochastic training (for general smooth losses) remain unexplored. Our main purpose in this work is to theoretically establish that full-batch GD indeed generalizes efficiently for general smooth losses. While SGD appears to generalize better than full-batch GD for non-smooth and Lipschitz convex losses (Bassily et al., 2020; Amir et al., 2021), non-smoothness seems to be problematic for efficient algorithmic generalization. In fact, tightness analysis on non-smooth losses (Bassily et al., 2020) shows that the generalization error bounds become vacuous for standard step-size choices. Our work shows that for general smooth losses, full-batch GD achieves either tighter stability and excess error rates (convex case), or equivalent rates (compared to SGD in the strongly convex setting) but with significantly shorter training horizon (strongly-convex objective).

## 2 RELATED WORK AND CONTRIBUTIONS

Let $n$ denote the number of available training samples (examples). Recent results (Lei & Tang, 2021; Zhou et al., 2022) on SGD provided bounds of order $\mathcal{O}(1/\sqrt{n})$ for Lipschitz and smooth nonconvex losses. Neu et al. (2021) also provided generalization bounds of order $\mathcal{O}(\eta T/\sqrt{n})$, with $T = \sqrt{n}$ and step-size $\eta = 1/T$ to recover the rate $\mathcal{O}(1/\sqrt{n})$. In contrast, we show that full-batch GD generalizes efficiently for appropriate choices of decreasing learning rate that guarantees faster convergence and smaller generalization error, simultaneously. Additionally, the generalization error involves an intrinsic dependence on the set of the stationary points and the initial point. Specifically,

| Full-Batch Gradient Descent | | |
|---|---|---|
| Step Size | Excess Risk | Loss |
| $\eta_t \le c/\beta t, \forall c < 1$ | $C\dfrac{T^c\sqrt{\log(T)+1}}{n} + \epsilon_{\text{opt}}$ | Nonconvex |
| $\eta_t = 1/2\beta$ | $C\left(\dfrac{T\epsilon_{\mathbf{c}}+1}{n} + \dfrac{1}{T}\right)$ | Convex |
| $\eta_t = 1/2\beta$, $T = \sqrt{n}$ | $C\dfrac{\epsilon_{\mathbf{c}}+1}{\sqrt{n}} + \mathcal{O}\left(\dfrac{1}{n}\right)$ | Convex |
| $\eta_t = 2/(\beta+\gamma)$ | $C\left(\sqrt{\epsilon_{\mathbf{c}}}\dfrac{\sqrt{T}}{n} + \exp\left(\dfrac{-4T\gamma}{\beta+\gamma}\right)\right) + \mathcal{O}\left(\dfrac{1}{n^2}\right)$ | $\gamma$-Strongly Convex |
| $\eta_t = 2/(\beta+\gamma)$ $T = \dfrac{(\beta+\gamma)\log n}{2\gamma}$ | $C\sqrt{\epsilon_{\mathbf{c}}}\dfrac{\sqrt{\log(n)}}{n} + \mathcal{O}\left(\dfrac{1}{n^2}\right)$ | $\gamma$-Strongly Convex |

Table 2: A list of excess risk bounds for the full-batch GD up to some constant factor $C > 0$. We denote the number of samples by $n$. "$\epsilon_{\text{opt}}$" denotes the optimization error $\epsilon_{\text{opt}} \triangleq \mathbb{E}[R_S(A(S)) - R_S^*]$, $T$ is the total number of iterations and $\epsilon_{\mathbf{c}} \triangleq \mathbb{E}[R_S(W_S^*)]$ is the model capacity (interpolation) error.

we show that full-batch GD with the decreasing learning rate choice of $\eta_t = 1/2\beta t$ achieves tighter bounds of the order $\mathcal{O}(\sqrt{T\log(T)}/n)$ (since $\sqrt{T\log(T)}/n \le 1/\sqrt{n}$) for any $T \le n/\log(n)$. In fact, $\mathcal{O}(\sqrt{T\log(T)}/n)$ essentially matches the rates in prior works (Hardt et al., 2016) for smooth and Lipschitz (and often bounded) loss, however we assume only smoothness at the expense of the $\sqrt{\log(T)}$ term. Further, for convex losses we show that full-batch GD attains tighter generalization error and excess risk bounds than those of SGD in prior works (Lei & Ying, 2020b), or similar rates in comparison with prior works that consider additional assumptions (Lipschitz or sub-Gaussian loss) (Hardt et al., 2016; Lugosi & Neu, 2022; Kozachkov et al., 2022). In fact, for convex losses and for a fixed step-size $\eta_t = 1/2\beta$, we show generalization error bounds of order $\mathcal{O}(T/n)$ for non-Lipschitz losses, while SGD bounds in prior work are of order $\mathcal{O}(\sqrt{T/n})$ (Lei & Ying, 2020b). As a consequence, full-batch GD attains improved generalization error rates by one order of error magnitude and appears to be more stable in the non-Lipschitz case, however tightness guarantees for non-Lipschitz losses remains an open problem.

Our results also establish that full-batch GD provably achieves efficient excess error rates through fewer number of iterations, as compared with the state-of-the-art excess error guarantees for SGD. Specifically, for convex losses with limited model capacity (non-interpolation), we show that with constant step size and $T = \sqrt{n}$, the excess risk is of the order $\mathcal{O}(1/\sqrt{n})$, while the SGD algorithm requires $T = n$ to achieve excess risk of the order $\mathcal{O}(1/\sqrt{n})$ (Lei & Ying, 2020b, Corollary 5.a).

For $\gamma$-strongly convex objectives, our analysis for full-batch GD relies on a *leave-one-out* $\gamma_{loo}$-strong convexity of the objective instead of the full loss function being strongly-convex. This property relaxes strong convexity, while it provides stability and generalization error guarantees that recover the convex loss setting when $\gamma_{loo} \to 0$. Prior work (Lei & Ying, 2020b, Section 6, Stability with Relaxed Strong Convexity) requires a Lipschitz loss, while the corresponding bound becomes infinity when $\gamma \to 0$, in contrast to the leave-one-out approach. Further, prior guarantees on SGD (Lei & Ying, 2020b, Theorem 11 and Theorem 12) often achieve the same rate of $\mathcal{O}(1/n)$, however with $T = \Theta(n)$ iterations (and a Lipschitz loss), in contrast with our full-batch GD bound that requires only $T = \Theta(\log n)$ iterations (at the expense of a $\sqrt{\log(n)}$ term)[1]. Finally, our approach does not require a projection step (in contrast to Hardt et al. (2016); Lei & Ying (2020b)) in the update rule and consequently avoids dependencies on possibly large Lipschitz constants.

---

[1]SGD naturally requires less computation than GD. However, the directional step of GD can be evaluated in parallel. As a consequence, for a strongly-convex objective GD would be more efficient than SGD (in terms of running time) if some parallel computation is available.

In summary, we show that for smooth nonconvex, convex and strongly convex losses, full-batch GD generalizes, which provides an explanation of its good empirical performance in practice (Hoffer et al., 2017; Geiping et al., 2022). We refer the reader to Table 1 for an overview and comparison of our excess risk bounds and those of prior work (on SGD). A more detailed presentation of the bounds appears in Table 2 (see also Appendix A, Table 3 and Table 4).

## 3 PROBLEM STATEMENT

Let $f(w, z)$ be the loss at the point $w \in \mathbb{R}^d$ for some example $z \in \mathcal{Z}$. Given a dataset $S \triangleq \{z_i\}_{i=1}^n$ of i.i.d samples $z_i$ from an unknown distribution $\mathcal{D}$, our goal is to find the parameters $w^*$ of a learning model such that $w^* \in \arg\min_w R(w)$, where $R(w) \triangleq \mathbb{E}_{Z \sim \mathcal{D}}[f(w, Z)]$ and $R^* \triangleq R(w^*)$. Since the distribution $\mathcal{D}$ is not known, we consider the empirical risk

$$R_S(w) \triangleq \frac{1}{n} \sum_{i=1}^n f(w; z_i). \tag{1}$$

The corresponding empirical risk minimization (ERM) problem is to find $W_S^* \in \arg\min_w R_S(w)$ (assuming minimizers on data exist for simplicity) and we define $R_S^* \triangleq R_S(W_S^*)$. For a deterministic algorithm $A$ with input $S$ and output $A(S)$, the excess risk $\epsilon_{\text{excess}}$ is bounded by the sum of the generalization error $\epsilon_{\text{gen}}$ and the optimization error $\epsilon_{\text{opt}}$ (Hardt et al., 2016, Lemma 5.1), (Dentcheva & Lin, 2022)

$$\epsilon_{\text{excess}} \triangleq \mathbb{E}[R(A(S))] - R^* \leq \underbrace{\mathbb{E}[R(A(S)) - R_S(A(S))]}_{\epsilon_{\text{gen}}} + \underbrace{\mathbb{E}[R_S(A(S))] - \mathbb{E}[R_S(W_S^*)]}_{\epsilon_{\text{opt}}}. \tag{2}$$

For the rest of the paper we assume that the loss is smooth and non-negative. These are the only globally required assumptions on the loss function.

**Assumption 1 ($\beta$-Smooth Loss)** *The gradient of the loss function is $\beta$-Lipschitz*

$$\|\nabla_w f(w, z) - \nabla_u f(u, z)\|_2 \leq \beta \|w - u\|_2, \quad \forall z \in \mathcal{Z}. \tag{3}$$

Additionally, we define the interpolation error that will also appear in our results.

**Definition 1 (Model Capacity/Interpolation Error)** *Define $\epsilon_{\mathbf{c}} \triangleq \mathbb{E}[R_S(W_S^*)]$.*

In general $\epsilon_{\mathbf{c}} \geq 0$ (non-negative loss). If the model has sufficiently large capacity, then for almost every $S \in \mathbb{Z}^n$, it is true that $R_S(W_S^*) = 0$. Equivalently, it holds that $\epsilon_{\mathbf{c}} = 0$. In the next section we provide a general theorem for the generalization error that holds for any symmetric deterministic algorithm (e.g. full-batch gradient descent) and any smooth loss under memorization of the data-set.

## 4 SYMMETRIC ALGORITHM AND SMOOTH LOSS

Consider the i.i.d random variables $z_1, z_2, \ldots, z_n, z_1', z_2', \ldots, z_n'$, with respect to an unknown distribution $\mathcal{D}$, the sets $S \triangleq (z_1, z_2, \ldots, z_n)$ and $S^{(i)} \triangleq (z_1, z_2, \ldots, z_i', \ldots, z_n)$ that differ at the $i^{\text{th}}$ random element. Recall that an algorithm is symmetric if the output remains unchanged under permutations of the input vector. Then (Bousquet & Elisseeff, 2002, Lemma 7) shows that for any $i \in \{1, \ldots, n\}$ and any symmetric deterministic algorithm $A$ the generalization error is $\epsilon_{\text{gen}} = \mathbb{E}_{S^{(i)}, z_i}[f(A(S^{(i)}); z_i) - f(A(S); z_i)]$. Identically, we write $\epsilon_{\text{gen}} = \mathbb{E}[f(A(S^{(i)}); z_i) - f(A(S); z_i)]$, where the expectation is over the random variables $z_1, \ldots, z_n, z_1', \ldots, z_n'$ for the rest of the paper. We define the model parameters $W_t, W_t^{(i)}$ evaluated at time $t$ with corresponding inputs $S$ and $S^{(i)}$. For brevity, we also provide the next definition.

**Definition 2** *We define the* expected output stability *as $\epsilon_{\text{stab}(A)} \triangleq \mathbb{E}[\|A(S) - A(S^{(i)})\|_2^2]$ and the* expected optimization error *as $\epsilon_{\text{opt}} \triangleq \mathbb{E}[R_S(A(S)) - R_S(W_S^*)]$.*

We continue by providing an upper bound that connects the generalization error with the expected output stability and the expected optimization error at the final iterate of the algorithm.

**Theorem 3 (Generalization Error)** *Let $f(\cdot\,;z)$ be non-negative $\beta$-smooth loss for any $z \in \mathcal{Z}$. For any symmetric deterministic algorithm $A(\cdot)$ the generalization error is bounded as*

$$|\epsilon_{\mathrm{gen}}| \leq 2\sqrt{2\beta(\epsilon_{\mathrm{opt}} + \epsilon_{\mathbf{c}})\epsilon_{\mathrm{stab}(A)}} + 2\beta\epsilon_{\mathrm{stab}(A)}, \tag{4}$$

*where $\epsilon_{\mathrm{stab}(A)} \triangleq \mathbb{E}[\|A(S) - A(S^{(i)})\|_2^2]$. In the limited model capacity case it is true that $\epsilon_{\mathbf{c}}$ is positive (and independent of $n$ and $T$) and $|\epsilon_{\mathrm{gen}}| = \mathcal{O}(\sqrt{\epsilon_{\mathrm{stab}(A)}})$.*

We provide the proof of Theorem 3 in Appendix B.1. The generalization error bound in equation 4 holds for any symmetric algorithm and smooth loss. Theorem 3 consist the tightest variant of (Lei & Ying, 2020b, Theorem 2, b)) and shows that the expected output stability and a small expected optimization error at termination sufficiently provide an upper bound on the generalization error for smooth (possibly non-Lipschitz) losses. Further, the optimization error term $\epsilon_{\mathrm{opt}}$ is always bounded and goes to zero (with specific known rates) in the cases of (strongly) convex losses. Under the interpolation condition the generalization error bound satisfies tighter bounds.

**Corollary 4 (Generalization under Memorization)** *If memorization of the training set is feasible under sufficiently large model capacity, then $\epsilon_{\mathbf{c}} = 0$ and consequently $|\epsilon_{\mathrm{gen}}| \leq 2\sqrt{2\beta\epsilon_{\mathrm{opt}}\epsilon_{\mathrm{stab}(A)}} + 2\beta\epsilon_{\mathrm{stab}(A)}$ and $|\epsilon_{\mathrm{gen}}| = \mathcal{O}(\max\{\sqrt{\epsilon_{\mathrm{opt}}\epsilon_{\mathrm{stab}(A)}}, \epsilon_{\mathrm{stab}(A)}\})$.*

For a small number of iterations $T$ the above error rate is equivalent with Theorem 3. For sufficiently large $T$ the optimization error rate matches the expected output stability and provides a tighter rate (with respect to that of Theorem 3) of $|\epsilon_{\mathrm{gen}}| = \mathcal{O}(\epsilon_{\mathrm{stab}(A)})$.

**Remark.** As a byproduct of Theorem 3, one can show generalization and excess risk bounds for a uniformly $\mu$-PL objective (Karimi et al., 2016) defined as $\mathbb{E}[\|\nabla R_S(w)\|_2^2] \geq 2\mu\mathbb{E}[R_S(w) - R_S^*]$ for all $w \in \mathbb{R}^d$. Let $\pi_S \triangleq \pi(A(S))$ be the projection of the point $A(S)$ to the set of the minimizers of $R_S(\cdot)$. Further, define the constant $\tilde{c} \triangleq \mathbb{E}[R_S(\pi_S) + R(\pi_S)]$. Then a bound on the excess risk is (the proof appears in Appendix D)

$$\epsilon_{\mathrm{excess}} \leq \frac{8\beta\sqrt{\tilde{c}}}{n\mu}\sqrt{\epsilon_{\mathrm{opt}} + \epsilon_{\mathbf{c}}} + \frac{8\sqrt{2\beta\epsilon_{\mathrm{opt}}\epsilon_{\mathbf{c}}}}{\sqrt{\mu}} + \frac{16\tilde{c}\beta^2}{n^2\mu^2} + \frac{45\beta}{\mu}\epsilon_{\mathrm{opt}}. \tag{5}$$

We note that a closely related bound has been shown in (Lei & Ying, 2020a). In fact, (Lei & Ying, 2020a, Therem 7) requires simultaneously the interpolation error to be zero ($\epsilon_{\mathbf{c}} = 0$) and an additional assumption, namely the inequality $\beta \leq n\mu/4$, to hold. However, if $\beta \leq n\mu/4$ and $\epsilon_{\mathbf{c}} = 0$ (interpolation assumption), then (Lei & Ying, 2020a, inequality (B.13), Proof of Theorem 1) implies that ($\mathbb{E}[R(\pi_S)] \leq 3\mathbb{E}[R_S(\pi_S)]$ and) the expected *population* risk at $\pi_S$ is zero, i.e., $\mathbb{E}[R(\pi_S)] = 0$. Such a situation is apparently trivial since the population risk is zero at the empirical minimizer $\pi_S \in \arg\min R_S(\cdot)$.[2] On the other hand, if $\beta > n\mu/4$, these bounds become vacuous. A PL condition is interesting under the interpolation regime and since the PL is not uniform (with respect to the data-set) in practice (Liu et al., 2022), it is reasonable to consider similar bounds to that of 5 as trivial.

## 5 FULL-BATCH GD

In this section, we derive generalization error and excess risk bounds for the full-batch GD algorithm. We start by providing the definition of the expected path error $\epsilon_{\mathrm{path}}$, in addition to the optimization error $\epsilon_{\mathrm{opt}}$. These quantities will prominently appear in our analysis and results.

**Definition 5 (Path Error)** *For any $\beta$-smooth (possibly) nonconvex loss, learning rate $\eta_t$, and for any $i \in \{1, \ldots, n\}$, we define the expected path error as*

$$\epsilon_{\mathrm{path}} \triangleq \sum_{t=1}^{T} \eta_t \mathbb{E}[\|\nabla f(W_t, z_i)\|_2^2]. \tag{6}$$

---

[2]In general, this occurs when $n \to \infty$, and the generalization error is zero. As a consequence, the excess risk becomes equals to the optimization error (see also (Lei & Ying, 2020a, Therem 7)), and the analysis becomes not interesting from generalization error prospective.

The $\epsilon_{\text{path}}$ term expresses the path-dependent quantity that appears in the generalization bounds in our results[3]. Additionally, as we show, the generalization error also depends on the average optimization error $\epsilon_{\text{opt}}$ (Theorem 3). A consequence of the dependence on $\epsilon_{\text{opt}}$, is that full-batch GD generalizes when it reaches the neighborhoods of the loss minima. Essentially, the expected path error and optimization error replace bounds in prior works (Hardt et al., 2016; Kozachkov et al., 2022) that require a Lipschitz loss assumption to upper bound the gradients and substitute the Lipschitz constant with tighter quantities. Later we show the dependence of the expected output stability term in Theorem 3 with respect to the expected path error. Then through explicit rates for both $\epsilon_{\text{path}}$ and $\epsilon_{\text{opt}}$ we characterize the generalization error and excess risk.

## 5.1 NONCONVEX LOSS

We proceed with the average output stability and generalization error bounds for nonconvex smooth losses. Through a stability error bound, the next result connects Theorem 3 with the expected path error and the corresponding learning rate. Then we use that expression to derive generalization error bounds for the full-batch GD in the case of nonconvex losses.

**Theorem 6 (Stability Error — Nonconvex Loss)** *Assume that the (possibly) nonconvex loss $f(\cdot, z)$ is $\beta$-smooth for all $z \in \mathcal{Z}$. Consider the full-batch GD where $T$ denotes the total number of iterates and $\eta_t$ denotes the learning rate, for all $t \leq T + 1$. Then for the outputs of the algorithm $W_{T+1} \equiv A(S)$, $W_{T+1}^{(i)} \equiv A(S^{(i)})$ it is true that*

$$\epsilon_{\text{stab}(A)} \leq \frac{4\epsilon_{\text{path}}}{n^2} \sum_{t=1}^{T} \eta_t \prod_{j=t+1}^{T} \left(1 + \beta\eta_j\right)^2. \tag{7}$$

The expected output stability in Theorem 6 is bounded by the product of the expected path error (Definition 5), a sum-product term ($\sum_{t=1}^{T} \eta_t \prod_{j=t+1}^{T} \left(1 + \beta\eta_j\right)^2$) that only depends on the step-size and the term $4/n^2$ that provides the dependence on the sample complexity. In light of Theorem 3, and Theorem 6, we derive the generalization error of full-batch GD for smooth nonconvex losses.

**Theorem 7 (Generalization Error — Nonconvex Loss)** *Assume that the loss $f(\cdot, z)$ is $\beta$-smooth for all $z \in \mathcal{Z}$. Consider the full-batch GD where $T$ denotes the total number of iterates, and the learning rate is chosen as $\eta_t \leq C/t \leq 1/\beta$, for all $t \leq T + 1$. Let $\epsilon \triangleq \beta C < 1$ and $\bar{C}(\epsilon, T) \triangleq \min\{\epsilon + 1/2, \epsilon \log(eT)\}$. Then the generalization error of full-batch GD is bounded by*

$$|\epsilon_{\text{gen}}| \leq \frac{4\sqrt{2}}{n} \sqrt{(\epsilon_{\text{opt}} + \epsilon_{\mathbf{c}})\epsilon_{\text{path}}} (eT)^{\epsilon} \bar{C}^{\frac{1}{2}}(\epsilon, T) + 8\frac{\epsilon_{\text{path}}}{n^2}(eT)^{2\epsilon}\bar{C}(\epsilon, T)$$

$$\leq \frac{4\sqrt{3}(eT)^{\epsilon}}{n} \sqrt{(\epsilon_{\text{opt}} + \epsilon_{\mathbf{c}})\epsilon_{\text{path}}} + 12\frac{(eT)^{2\epsilon}}{n^2}\epsilon_{\text{path}}. \tag{8}$$

Additionally, by the definition of the expected path and optimization error, and from the descent direction of algorithm, we evaluate upper bounds on the terms $\epsilon_{\text{path}}$ and $\epsilon_{\text{opt}}$ and derive the next bound as a byproduct of Theorem 7.

**Corollary 8** *The generalization error of full-batch GD in Theorem 7 can be further bounded as*

$$|\epsilon_{\text{gen}}| \leq \left(\frac{8\sqrt{3}}{n}\sqrt{\log(eT)}(eT)^{\epsilon} + \frac{48}{n^2}\log(eT)(eT)^{2\epsilon}\right) \mathbb{E}[R_S(W_1)]. \tag{9}$$

The inequality (8) in Theorem 7 shows the explicit dependence of the generalization error bound on the path-dependent error $\epsilon_{\text{path}}$ and the optimization error $\epsilon_{\text{opt}}$. Note that during the training process the path-dependent error increases, and the optimization error decreases. Both terms $\epsilon_{\text{path}}$ and $\epsilon_{\text{opt}}$ may be upper bounded, to find the simplified (but potentially looser) bound appeared in Corollary 8. We prove Theorem 6, Theorem 7 and Corollary 8 in Appendix C. Finally, the generalization error in Corollary 8 matches bounds in prior work, including information theoretic bounds for the SGLD algorithm (Wang et al., 2021b, Corollary 1) (with fixed step-size), while our results do not require the sub-Gaussian loss assumption and show that similar generalization is achievable through deterministic training.

---

[3]Recall that the initial point $W_1$ may be chosen arbitrarily and uniformly over the dataset.

**Remark. (Dependence on Stationary Points)** Let $W_1$ be an arbitrary initial point (independent of $S$). Under mild assumptions (provided in (Lee et al., 2016)) GD convergences to (local) minimizers. Let $W^*_{S,W_1}$ be the stationary point such $\lim_{T\uparrow\infty} A(S) \to W^*_{S,W_1}$. Then through the smoothness of the loss, we derive an alternative form of the generalization error bound in Theorem 3 that expresses the dependence of the generalization error with respect to the quality of the set of stationary points, i.e.,

$$|\epsilon_{\text{gen}}| \leq 4\sqrt{\beta\left(\beta\mathbb{E}[\|A(S) - W^*_{S,W_1}\|^2_2] + \mathbb{E}[R_S(W^*_{S,W_1})]\right)\epsilon_{\text{stab}(A)}} + 2\beta\epsilon_{\text{stab}(A)} \quad (10)$$

Inequality (10) provides a detailed bound that depends on the expected loss at the stationary point and the expected distance of the output from the stationary point, namely $\mathbb{E}[\|A(S) - W^*_{S,W_1}\|^2_2]$.

## 5.2 CONVEX LOSS

In this section, we provide generalization error guarantees for GD on smooth convex losses. Starting from the stability of the output of the algorithm, we show that the dependence on the learning rate is weaker than that of the nonconvex case. That dependence and the fast convergence to the minimum guarantee tighter generalization error bounds than the general case of nonconvex losses in Section 5.1. The generalization error and the corresponding optimization error bounds provide an excess risk bound through the error decomposition (2). We refer the reader to Table 2 for a summary of the excess risk guarantees. We continue by providing the stability bound for convex losses.

**Theorem 9 (Stability Error — Convex Loss)** *Assume that the convex loss $f(\cdot, z)$ is $\beta$-smooth for all $z \in \mathcal{Z}$. Consider the full-batch GD where $T$ denotes the total number of iterates and $\eta_t \leq 1/2\beta$ learning rate, for all $t \leq T + 1$. Then for outputs of the algorithm $W_{T+1} \equiv A(S), W^{(i)}_{T+1} \equiv A(S^{(i)})$ it is true that*

$$\epsilon_{\text{stab}(A)} \leq \frac{4\epsilon_{\text{path}}}{n^2} \sum_{t=1}^{T} \eta_t \leq \frac{32\beta \sum_{t=1}^{T} \eta_t}{n^2}\left(\mathbb{E}[\|W_1 - W^*_S\|^2_2] + \epsilon_{\mathbf{c}} \sum_{t=1}^{T} \eta_t\right) \quad (11)$$

In the convex case, the expected output stability (inequality 11) is bounded by the product of the expected path error, the number of samples term $2/n^2$ and the accumulated learning rate. The inequality (11) gives $\epsilon_{\text{stab}(A)} = \mathcal{O}((\sum_{t=1}^{T} \eta_t/n)^2)$ and through Theorem 3 we find $|\epsilon_{\text{gen}}| = \mathcal{O}(\sum_{t=1}^{T} \eta_t/n)$. In contrast, stability guarantees for the SGD and non-Lipschitz losses in prior work (Lei & Ying, 2020b, Theorem 3, (4.4)) give $\epsilon_{\text{stab}(A)} = \mathcal{O}\left(\sum_{t=1}^{T} \eta_t^2/n\right)$ and $|\epsilon_{\text{gen}}| = \mathcal{O}(\sqrt{\sum_{t=1}^{T} \eta_t^2/n})$. As a consequence, GD guarantees are tighter than existing bounds of the SGD for non-Lipschitz losses, a variety of learning rates and $T \leq n$. For instance, for fixed $\eta_t = 1/\sqrt{T}$, the generalization error bound of GD is $|\epsilon_{\text{gen}}| = \mathcal{O}(\sqrt{T}/n)$ which is tighter than the corresponding bound of SGD, namely $|\epsilon_{\text{gen}}| = \mathcal{O}(1/\sqrt{n})$. Further, GD applies for much larger learning rates ($\eta_t = 1/\beta$), which provide not only tighter generalization error bound but also tighter excess risk guarantees than SGD as we later show. By combining Theorem 3 and Theorem 9, we show the next generalization error bound.

**Theorem 10 (Generalization Error — Convex Loss)** *Let the loss function $f(\cdot, z)$ be convex and $\beta$-smooth for all $z \in \mathcal{Z}$. Consider the full-batch GD where $T$ denotes the total number of iterates. We chose the learning rate such that $\eta_t \leq 1/2\beta$, for all $t \leq T + 1$. Then the generalization error of full-batch GD is bounded by*

$$|\epsilon_{\text{gen}}| \leq \frac{4\sqrt{2\beta\left(\epsilon_{\text{opt}} + \epsilon_{\mathbf{c}}\right)\epsilon_{\text{path}}}}{n}\sqrt{\sum_{t=1}^{T}\eta_t} + 8\beta\frac{\epsilon_{\text{path}}}{n^2}\sum_{t=1}^{T}\eta_t. \quad (12)$$

We provide the proof of Theorem 9 and Theorem 10 in Appendix E. Similar to the nonconvex case (Theorem 7), the bound in Theorem 10 shows the explicit dependence of the generalization error on the number of samples $n$, the path-dependent term $\epsilon_{\text{path}}$, and the optimization error $\epsilon_{\text{opt}}$, as well as the effect of the accumulated learning rate. From the inequality (12), we can proceed by deriving exact bounds on the optimization error and the accumulated learning rate, to find explicit expressions of the generalization error bound. Through Theorem 9, Theorem 10 (and Lemma 20 in Appendix E), we derive explicit generalization error bounds for certain choices of the learning rate. In fact, we consider the standard choice $\eta_t = 1/2\beta$ in the next result.

**Theorem 11 (Generalization/Excess Error — Convex Loss)** *Let the loss function $f(\cdot, z)$ be convex and $\beta$-smooth for all $z \in \mathcal{Z}$. If $\eta_t = 1/2\beta$ for all $t \in \{1, \ldots, T\}$, then*

$$|\epsilon_{\text{gen}}| \leq 8 \left( \frac{1}{n} + \frac{2T}{n^2} \right) \left( 3\beta \mathbb{E}[\|W_1 - W_S^*\|_2^2] + T\epsilon_{\mathbf{c}} \right), \tag{13}$$

*and*

$$\epsilon_{\text{excess}} \leq 8 \left( \frac{1}{n} + \frac{2T}{n^2} \right) \left( 3\beta \mathbb{E}[\|W_1 - W_S^*\|_2^2] + T\epsilon_{\mathbf{c}} \right) + \frac{3\beta \mathbb{E}[\|W_1 - W_S^*\|_2^2]}{T}. \tag{14}$$

As a consequence, for $T = \sqrt{n}$ iterations the GD algorithm achieves $\epsilon_{\text{excess}} = \mathcal{O}(1/\sqrt{n})$. In contrast, SGD requires $T = n$ number of iterations to achieve $\epsilon_{\text{excess}} = \mathcal{O}(1/\sqrt{n})$ (Lei & Ying, 2020b, Corollary 5, a)). However, if $\epsilon_{\mathbf{c}} = 0$, then both algorithms have the same excess risk rate of $\mathcal{O}(1/n)$ through longer training with $T = n$ iterations. Finally, observe that the term $\mathbb{E}[\|W_1 - W_S^*\|]_2^2$ should be $\mathcal{O}(1)$ and independent of the parameters of interest (for instance $n$) to derive the aforementioned rates.

## 5.3 STRONGLY-CONVEX OBJECTIVE

One common approach to enforce strong-convexity is through explicit regularization. In such a case both the objective $R_S(\cdot)$ the individual losses $f(\cdot; z)$ are strongly-convex. In other practical scenarios, the objective is often strongly-convex but the individual losses are not (Ma et al., 2018, Section 3). In this section, we show stability and generalization error guarantees that include the above cases by assuming a $\gamma$-strongly convex objective. We also show a property of full-batch GD that requires only a *leave-one-out* variant of the objective to be strongly-convex. If the objective $R_S(\cdot)$ is $\gamma$-strongly convex and the loss $f(\cdot; z)$ is $\beta$-smooth, then the leave-one-out function $R_{S^{-i}}(w) \triangleq \sum_{j=1, j \neq i}^n f(w; z_j)/n$ is $\gamma_{loo}$-strongly convex for all $i \in \{1, \ldots, n\}$ for some $\gamma_{loo} \leq \gamma$. Although $\gamma_{loo}$ is slightly smaller than $\gamma$ ($\gamma_{loo} = \max\{\gamma - \beta/n, 0\}$), our results reduce to the convex loss generalization and stability bounds when $\gamma_{loo} \to 0$. Further, the faster convergence also provides tighter bounds for the excess risk (see Table 2).

**Theorem 12 (Stability Error — Strongly Convex Loss)** *Assume that the loss $f(\cdot, z)$ is $\beta$-smooth for all $z \in \mathcal{Z}$ and that $R_S(\cdot)$ is $\gamma$-strongly convex. Consider the full-batch GD where $T$ denotes the total number of iterates and $\eta_t \leq 2/(\beta + \gamma)$ denotes the learning rate, for all $t \leq T$. Then for outputs of the algorithm $W_{T+1} \equiv A(S)$, $W_{T+1}^{(i)} \equiv A(S^{(i)})$ it is true that*

$$\epsilon_{\text{stab}(A)} \leq \frac{4\epsilon_{\text{path}}}{n^2} \sum_{t=1}^T \eta_t \prod_{j=t+1}^T (1 - \eta_j \gamma_{loo}). \tag{15}$$

*Specifically, if $\eta_t = 2/(\beta + \gamma)$, then*

$$\epsilon_{\text{stab}(A)} \leq \frac{4\epsilon_{\text{path}}}{n^2} \min \left\{ \frac{1}{\gamma_{loo}}, \frac{2T}{\beta} \right\}. \tag{16}$$

By comparing the stability guarantee of Theorem 9 with Theorem 12, we observe that the learning rate dependent term (sum-product) is smaller than that of the convex case. While the dependence on expected path error ($\epsilon_{\text{path}}$) is identical, we show (Appendix F) that the $\epsilon_{\text{path}}$ term is smaller in the strongly convex case. Additionally, Theorem 12 recovers the stability bounds of the convex loss case, when $\gamma_{loo} \to 0$ (and possibly $\gamma \to 0$). Similarly to the nonconvex and convex loss cases, Theorem 3 and the stability error bound in Theorem 12 provide the generalization error bound for strongly convex losses.

**Theorem 13 (Generalization Error — Strongly Convex Loss)** *Let the loss function $f(\cdot, z)$ $\beta$-smooth for all $z \in \mathcal{Z}$ and the objective $R_S(\cdot)$ be $\gamma$-strongly convex. Consider the full-batch GD where $T$ denotes the total number of iterates. Let us set the learning rate to $\eta_t \leq 2/(\beta + \gamma)$, for all $t \leq T$. Then the generalization error of full-batch GD is bounded by*

$$|\epsilon_{\text{gen}}| \leq \frac{4\sqrt{(\epsilon_{\text{opt}} + \epsilon_{\mathbf{c}})\epsilon_{\text{path}}}}{n} \sqrt{2\beta \sum_{t=1}^T \eta_t \prod_{j=t+1}^T (1 - \eta_j \gamma_{loo}) + 8\beta \frac{\epsilon_{\text{path}}}{n^2} \sum_{t=1}^T \eta_t \prod_{j=t+1}^T (1 - \eta_j \gamma_{loo})}.$$

We prove Theorem 12 and Theorem 13 in Appendix F. Recall that the sum-product term in the inequality of Theorem 13 is smaller than the summation of the learning rates in Theorem 10. This fact together with the tighter optimization error bound provide a smaller excess risk than those of the convex losses. Similar to the convex loss setting, we use known optimization error guarantees of full-batch GD for strongly convex losses to derive explicit expressions of the generalization and excess risk bounds. By combining Theorem 13 and optimization and path error bounds (Lemma 22, Lemma 23 in Appendix F, and Lemma 15 in Appendix B.2), we derive our generalization error bound for fixed step size as follows in the next result.

**Theorem 14 (Generalization/Excess — Strongly Convex Loss)** *Let the objective function $R_S(\cdot)$ be $\gamma$-strongly convex and $\beta$-smooth by choosing some $\beta$-smooth loss $f(\cdot, z))$, not necessarily (strongly) convex for all $z \in \mathcal{Z}$. Define $m(\gamma_{loo}, T) \triangleq \beta T \min \{\beta/\gamma_{loo}, 2T\}/(\beta + \gamma)$ and $M(W_1) \triangleq \max \{\beta \mathbb{E}[\|W_1 - W_S^*\|_2^2], \mathbb{E}[R_S(W_S^*)]\}$, and set the learning rate to $\eta_t = 2/(\beta + \gamma)$. Then the generalization error of the full-batch GD at the last iteration satisfies the inequality*

$$|\epsilon_{\text{gen}}| \leq \frac{8\sqrt{6}}{n} \left( \sqrt{M(W_1)} + \left( \exp \left( \frac{-2T\gamma}{\beta + \gamma} \right) + \frac{4\sqrt{3}}{n} \sqrt{m(\gamma_{loo}, T)} \right) M(W_1) \right) \sqrt{m(\gamma_{loo}, T)}.$$

*Additionally the optimization error (Lemma 23 in Appendix F) and the inequality (2) give the following excess risk*

$$\epsilon_{\text{excess}} \leq \frac{8\sqrt{6}}{n} \left[ \sqrt{\Delta_T} + \left( \exp \left( \frac{-2T\gamma}{\beta + \gamma} \right) + \frac{4\sqrt{3}}{n} \Delta_T \right) \right] + \Lambda \exp \left( \frac{-4T\gamma}{\beta + \gamma} \right), \qquad (17)$$

*where $\Delta_T \triangleq \beta T M(W_1) \min \{\beta/\gamma_{loo}, 2T\}/(\beta + \gamma)$ and $\Lambda \triangleq \beta \mathbb{E}[\|W_1 - W_S^*\|_2^2]/2$.*

Theorem 13 and Theorem 14 also recover the convex setting when $\gamma \to 0$ or $\gamma_{loo} \to 0$. Additionally, for $\gamma > 0$ and by setting the number of iterations as $T = (\beta/\gamma + 1) \log(n)/2$ and by defining the sequence $m_{n, \gamma_{loo}} \triangleq \beta \min \{\beta/\gamma_{loo}, (\beta/\gamma + 1) \log n\}/2\gamma$, the last inequality gives

$$|\epsilon_{\text{gen}}| \leq \frac{8\sqrt{6 \log n}}{n} \left( \sqrt{M(W_1)} + \frac{1 + 4\sqrt{3 m_{n, \gamma_{loo}}}}{n} M(W_1) \right) \sqrt{m_{n, \gamma_{loo}}}. \qquad (18)$$

Finally, for $T = (\beta/\gamma + 1) \log(n)/2$ iterations it is true that

$$\epsilon_{\text{excess}} \leq \frac{8\sqrt{6 \log n}}{n} \left( \sqrt{\Gamma_n} + \frac{1 + 4\sqrt{3}}{n} \Gamma_n \right) + \mathcal{O} \left( \frac{1}{n^2} \right), \qquad (19)$$

where $\Gamma_n \triangleq \beta M(W_1) \min \{\beta/\gamma_{loo}, (\beta/\gamma + 1) \log n\}/2\gamma$ and as a consequence the excess risk is of the order $\mathcal{O} \left( \sqrt{\log(n)}/n \right)$. As a comparison, the SGD algorithm (Lei & Ying, 2020b, Theorem 12) requires $T = n$ number of iterations to achieve an excess risk of the order $\mathcal{O}(1/n)$, while full-batch GD achieves essentially the same rate with $T = (\beta/\gamma + 1) \log(n)/2$ iterations.

# 6 CONCLUSION

In this paper we developed generalization error and excess risk guarantees for deterministic training on smooth losses via the the full-batch GD algorithm. At the heart of our analysis is a sufficient condition for generalization, implying that, for every symmetric algorithm, average algorithmic output stability and a small expected optimization error at termination ensure generalization. By exploiting this sufficient condition, we explicitly characterized the generalization error in terms of the number of samples, the learning rate, the number of iterations, a path-dependent quantity and the optimization error at termination, further exploring the generalization ability of full-batch GD for different types of loss functions. More specifically, we derived explicit rates on the generalization error and excess risk for nonconvex, convex and strongly convex smooth (possibly non-Lipschitz) losses/objectives. Our theoretical results shed light on recent empirical observations indicating that full-batch gradient descent generalizes efficiently and that stochastic training procedures might not be necessary and in certain cases may even lead to higher generalization errors and excess risks.

# 7 ACKNOWLEDGMENTS AND DISCLOSURE OF FUNDING

We would like to thank the four anonymous reviewers for providing valuable comments and suggestions, which have improved the presentation of the results and the overall quality of our paper.

Amin Karbasi acknowledges funding in direct support of this work from NSF (IIS-1845032), ONR (N00014- 19-1-2406), and the AI Institute for Learning-Enabled Optimization at Scale (TILOS).

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

| Full-Batch Gradient Descent | | |
|---|---|---|
| Step Size | Generalization Error | Loss |
| $\eta_t \leq C/\beta t, \forall C < 1$ | $\dfrac{4e\sqrt{3}}{n}T^C\sqrt{(\epsilon_{\text{opt}}+\epsilon_{\mathbf{c}})\epsilon_{\text{path}}} + \dfrac{12e^2}{n^2}T^{2C}\epsilon_{\text{path}}$ | NC |
| $\eta_t \leq C/\beta t, \forall C < 1$ | $48\left(\dfrac{\sqrt{\log(eT)}(eT)^C}{n} + \dfrac{\log(eT)(eT)^{2C}}{n^2}\right)\mathbb{E}[R_S(W_1)]$ | NC |
| $\eta_t = 1/2\beta$ | $8\left(\dfrac{1}{n} + \dfrac{2T}{n^2}\right)\left(3\beta\mathbb{E}[\|W_1 - W_S^*\|_2^2] + T\epsilon_{\mathbf{c}}\right)$ | C |
| $\eta_t = 2/(\beta+\gamma)$ | $\dfrac{8\sqrt{6}}{n}\left[\sqrt{\Delta_T} + \left(\exp\left(\dfrac{-2T\gamma}{\beta+\gamma}\right) + \dfrac{4\sqrt{3}}{n}\Delta_T\right)\right]$ | $\gamma$-SC |

Table 3: A list of the generalization error bounds for the full-batch GD. We denote the number of samples by $n$. $W_1$ is the initial point of the algorithm, and $W_S^*$ is a point in the set of minimizers of the objective. Also, "$\epsilon_{\text{path}}$" denotes the expected path error $\epsilon_{\text{path}} \triangleq \sum_{t=1}^T \eta_t \mathbb{E}[\|\nabla f(W_t, z_i)\|_2^2]$, "$\epsilon_{\text{opt}}$" denotes the optimization error $\epsilon_{\text{opt}} \triangleq \mathbb{E}[R_S(A(S)) - R_S^*]$, $T$ is the total number of iterations and we define the model capacity (interpolation) error risk as $\epsilon_{\mathbf{c}} \triangleq \mathbb{E}[R_S(W_S^*)]$. Lastly, we define the constant $\mathbf{M}(W_1) \triangleq \max\left\{\beta\mathbb{E}[\|W_1 - W_S^*\|_2^2], \mathbb{E}[R_S(W_S^*)]\right\}$ and the terms $\Gamma_n \triangleq \beta\mathbf{M}(W_1)\min\left\{\beta/\gamma_{loo}, (\beta/\gamma + 1)\log n\right\}/2\gamma$, $\Delta_T \triangleq \beta T M(W_1)\min\left\{\beta/\gamma_{loo}, 2T\right\}/(\beta+\gamma)$. Lastly, "NC", "C" and "$\gamma$-SC" correspond to nonconvex, convex and $\gamma$-strongly convex objective, respectively.

## A    SUMMARY OF THE RESULTS

Herein, we present a summary of the generalization and excess risk bounds. The detailed expressions of the generalization and excess risk bounds appear in Table 3 and 4.

## B    PROOFS

We provide the proofs of the results in these sections. We start by proving Theorem 3 and the bounds on the sum-product terms that appear in the stability error bounds, and then we continue with stability and generalization error guarantees, that we prove in parallel. We derive the excess risk bounds by applying the decomposition of the inequality (2).

### B.1    PROOF OF THEOREM 3

It is true that for any $i, j \in \{1, \ldots, n\}$

$$\mathbb{E}[f(A(S); z_i)] = \mathbb{E}[f(A(S); z_j)] = \frac{1}{n}\sum_{k=1}^n \mathbb{E}[f(A(S); z_k)] = \mathbb{E}[R_S(A(S))]. \qquad (20)$$

We show equation 20 through the symmetry of the algorithm (at each iteration) and the fact that $\{z_i\}_{i=1}^n$ are identically distributed as follows. The random variables $\{z_i\}_{i=1}^n$ remain exchangeable.[4] The $\beta$-smooth property of $f(\cdot; z)$ for all $z \in \mathcal{Z}$ gives

$$f(A(S^{(i)}); z) - f(A(S); z) \leq \langle A(S^{(i)}) - A(S), \nabla f(A(S); z)\rangle + \frac{\beta\|A(S^{(i)}) - A(S)\|_2^2}{2}. \qquad (21)$$

---

[4] $\mathbb{P}(z_1 = c_1, z_2 = c_2, \ldots, z_i = c_i, \ldots, z_j = c_j, \ldots, z_n = c_n, A(S) = \mathbf{w}) = \mathbb{P}(z_1 = c_1, z_2 = c_2, \ldots, z_i = c_j, \ldots, z_j = c_i, \ldots, z_n = c_n, A(S) = \mathbf{w})$ for any choice of the values $c_1, c_2, \ldots, c_n, \mathbf{w}$ and for any $i, j \in \{1, \ldots, n\}$.

| Full-Batch Gradient Descent | | |
|---|---|---|
| Step Size | Excess Risk | Loss |
| $\eta_t \leq C/\beta t,$ $\forall C < 1$ | $48 \left( \dfrac{\sqrt{\log(eT)}(eT)^C}{n} + \dfrac{\log(eT)(eT)^{2C}}{n^2} \right) \mathbb{E}[R_S(W_1)] + \epsilon_{\text{opt}}$ | NC |
| $\eta_t = 1/2\beta$ | $\left( \dfrac{8}{n} + \dfrac{16T}{n^2} \right) \left( 3\beta \mathbb{E}[\|W_1 - W_S^*\|_2^2] + T\epsilon_{\mathbf{c}} \right) + \dfrac{3\beta \mathbb{E}[\|W_1 - W_S^*\|_2^2]}{T}$ | C |
| $\eta_t = 1/2\beta,$ $T = \sqrt{n}$ | $8 \dfrac{\epsilon_{\mathbf{c}} + 3\beta \mathbb{E}[\|W_1 - W_S^*\|_2^2]}{\sqrt{n}} + \mathcal{O}\left( \dfrac{1}{n} \right)$ | C |
| $\eta_t = 2/(\beta + \gamma)$ | $\dfrac{8\sqrt{3}}{n} \left[ \sqrt{\Delta_T} + \left( \exp\left( \dfrac{-2T\gamma}{\beta + \gamma} \right) + \dfrac{4\sqrt{3}}{n} \Delta_T \right) \right] + \Lambda \exp\left( \dfrac{-4T\gamma}{\beta + \gamma} \right)$ | $\gamma$-SC |
| $\eta_t = 2/(\beta + \gamma)$ $T = \dfrac{(\beta+\gamma)\log n}{2\gamma}$ | $\dfrac{8\sqrt{3\log n}}{n} \left( \sqrt{\Gamma_n} + \dfrac{1 + 4\sqrt{3}}{n} \Gamma_n \right) + \mathcal{O}\left( \dfrac{1}{n^2} \right)$ | $\gamma$-SC |

Table 4: A list of excess risk bounds for the full-batch GD. We denote the number of samples by $n$. $W_1$ is the initial point of the algorithm, and $W_S^*$ is a point in the set of minimizers of the objective. Also, "$\epsilon_{\text{path}}$" denotes the expected path error $\epsilon_{\text{path}} \triangleq \sum_{t=1}^T \eta_t \mathbb{E}[\|\nabla f(W_t, z_i)\|_2^2]$, "$\epsilon_{\text{opt}}$" denotes the optimization error $\epsilon_{\text{opt}} \triangleq \mathbb{E}[R_S(A(S)) - R_S^*]$, $T$ is the total number of iterations and we define the model capacity (interpolation) error risk as $\epsilon_{\mathbf{c}} \triangleq \mathbb{E}[R_S(W_S^*)]$. Lastly, we define the constants $\Lambda \triangleq \beta \mathbb{E}[\|W_1 - W_S^*\|_2^2]/2$, $\mathbf{M}(W_1) \triangleq \max\left\{ \beta \mathbb{E}[\|W_1 - W_S^*\|_2^2], \mathbb{E}[R_S(W_S^*)] \right\}$ and the terms $\Gamma_n \triangleq \beta \mathbf{M}(W_1) \min\left\{ \beta/\gamma_{loo}, (\beta/\gamma + 1)\log n \right\}/2\gamma$, $\Delta_T \triangleq \beta T M(W_1) \min\left\{ \beta/\gamma_{loo}, 2T \right\}/(\beta + \gamma)$. Lastly, "NC", "C" and "$\gamma$-SC" correspond to nonconvex, convex and $\gamma$-strongly convex objective, respectively.

The expression $\epsilon_{\text{gen}} = \mathbb{E}[f(A(S^{(i)}); z_i) - f(A(S); z_i)]$ and the inequality (21) give

$$\epsilon_{\text{gen}} \leq \mathbb{E}\left[ \langle A(S^{(i)}) - A(S), \nabla f(A(S); z_i) \rangle + \frac{\beta \|A(S^{(i)}) - A(S)\|_2^2}{2} \right] \tag{22}$$

We find an upper bound for the expectation of the inner product of the inequality (22) by applying Cauchy-Schwartz inequality as

$$\mathbb{E}\left[ \langle A(S^{(i)}) - A(S), \nabla f(A(S); z_i) \rangle \right]$$
$$\leq \mathbb{E}\left[ \|A(S^{(i)}) - A(S)\|_2 \|\nabla f(A(S); z_i)\|_2 \right] \tag{23}$$
$$\leq \sqrt{\epsilon_{\text{stab}(A)} \mathbb{E}\left[ \|\nabla f(A(S); z_i)\|_2^2 \right]}, \tag{24}$$

here we use the inequalities $\langle a, b \rangle \leq \|a\|_2 \|b\|_2$ and $\mathbb{E}^2[XY] \leq \mathbb{E}[X^2]\mathbb{E}[Y^2]$ to derive the bounds in 23 and 24 respectively. By combining the inequalities 22 and 24 we find that for any $i \in \{1, \dots, n\}$ it is true that

$$\epsilon_{\text{gen}} \leq \sqrt{\epsilon_{\text{stab}(A)} \mathbb{E}\left[ \|\nabla f(A(S); z_i)\|_2^2 \right]} + \frac{\beta}{2} \epsilon_{\text{stab}(A)}. \tag{25}$$

To find an upper bound for the $|\epsilon_{\text{gen}}|$, we also need an upper bound for negative of $\epsilon_{\text{gen}}$, namely $\mathbb{E}[f(A(S); z_i) - f(A(S^{(i)}); z_i)] = -\epsilon_{\text{gen}}$. Note that by the same argument

$$-\epsilon_{\text{gen}} \leq \sqrt{\mathbb{E}\left[ \|A(S) - A(S^{(i)})\|_2^2 \right] \mathbb{E}\left[ \|\nabla f(A(S^{(i)}); z_i)\|_2^2 \right]} + \frac{\beta}{2} \mathbb{E}[\|A(S) - A(S^{(i)})\|_2^2]. \tag{26}$$

Then we find an upper bound on $\mathbb{E}[\|\nabla f(A(S^{(i)}); z_i)\|_2^2]$ as follows

$$\mathbb{E}\left[ \|\nabla f(A(S^{(i)}); z_i)\|_2^2 \right]$$

$$= \mathbb{E}\left[\|\nabla f(A(S^{(i)}); z_i) - \nabla f(A(S); z_i) + \nabla f(A(S); z_i)\|_2^2\right]$$

$$\leq 2\mathbb{E}\left[\|\nabla f(A(S^{(i)}); z_i) - \nabla f(A(S); z_i)]\|_2^2 + \|\nabla f(A(S); z_i)\|_2^2\right]$$

$$\leq 2\beta^2 \mathbb{E}[\|A(S) - A(S^{(i)})\|_2^2] + 2\mathbb{E}[\|\nabla f(A(S); z_i)]\|_2^2]. \tag{27}$$

The inequality 27 holds because of the $\beta$-smoothness of the loss. Additionally,

$$\sqrt{2\beta^2 \mathbb{E}^2\left[\|A(S) - A(S^{(i)})\|_2^2\right] + 2\mathbb{E}\left[\|A(S) - A(S^{(i)})\|_2^2\right]\mathbb{E}[\|\nabla f(A(S); z_i)\|_2^2]}$$

$$\leq \sqrt{2\mathbb{E}\left[\|A(S) - A(S^{(i)})\|_2^2\right]\mathbb{E}[\|\nabla f(A(S); z_i)\|_2^2]} + \sqrt{2}\beta\mathbb{E}[\|A(S) - A(S^{(i)})\|_2^2]. \tag{28}$$

We combine the inequalities 26, 27 and 28 to find

$$-\epsilon_{\text{gen}} \leq \sqrt{2\mathbb{E}\left[\|A(S) - A(S^{(i)})\|_2^2\right]\mathbb{E}[\|\nabla f(A(S); z_i)\|_2^2]} + 2\beta\mathbb{E}[\|A(S) - A(S^{(i)})\|_2^2]. \tag{29}$$

Finally, through the inequalities 25 and 29 we find

$$|\epsilon_{\text{gen}}| \leq \sqrt{2\epsilon_{\text{stab}(A)}\mathbb{E}[\|\nabla f(A(S); z_i)\|_2^2]} + 2\beta\epsilon_{\text{stab}(A)} \tag{30}$$

We use the self-bounding property of the non-negative $\beta$-smooth loss function $f(\cdot; z)$ (Srebro et al., 2010, Lemma 3.1), to show

$$\|\nabla f(A(S); z_i)\|_2^2 \leq 4\beta f(A(S); z_i). \tag{31}$$

The last display, Assumption 1 and equation 20 give

$$\mathbb{E}[\|\nabla f(A(S); z_i)\|_2^2] \leq 4\beta\mathbb{E}[f(A(S); z_i)] = 4\beta\frac{1}{n}\sum_{i=1}^{n}\mathbb{E}[f(A(S); z_i)]$$

$$= 4\beta\mathbb{E}[R_S(A(S))] \tag{32}$$

$$= 4\beta\left(\mathbb{E}[R_S(A(S))] - \mathbb{E}[R_S(W_S^*)] + \mathbb{E}[R_S(W_S^*)]\right)$$

$$= 4\beta\left(\epsilon_{\text{opt}} + \mathbb{E}[R_S(W_S^*)]\right). \tag{33}$$

We combine the inequalities 30, 33 and the Definition 1 to find

$$|\epsilon_{\text{gen}}| \leq 2\sqrt{2\beta\left(\epsilon_{\text{opt}} + \epsilon_{\mathbf{c}}\right)\epsilon_{\text{stab}(A)}} + 2\beta\epsilon_{\text{stab}(A)}. \tag{34}$$

The last inequality gives the bound on the generalization error and completes the proof. $\square$

## B.2  SUM PRODUCT TERMS IN THE STABILITY BOUNDS

Herein we show a lemma for the sum product terms associated with learning rate in Theorem 6 and Theorem 12. Then we will apply that lemma to derive the corresponding stability error bounds.

**Lemma 15** *The following are true:*

- *If $\eta_t = C \leq 2/(\beta + \gamma)$, then*

$$\sum_{t=1}^{T}\eta_t\prod_{j=t+1}^{T}(1 - \eta_j\gamma) = \frac{1 - (1 - C\gamma)^T}{\gamma}, \tag{35}$$

- *If $\eta_t = C/t \leq 2/(\beta + \gamma)$, for some $C \geq 2/\gamma$ for $t \geq 1 + \lceil\frac{\beta}{\gamma}\rceil$ and $\eta_t = C'/t \leq 2/(\beta + \gamma)$ for some $C' < 2/(\gamma + \beta)$ for $t \leq \lceil\frac{\beta}{\gamma}\rceil$, then*

$$\sum_{t=1}^{T}\eta_t\prod_{j=t+1}^{T}\left(1 - \frac{\eta_j\gamma}{2}\right) \leq C\log\left(e^2\lceil\beta/\gamma\rceil\right). \tag{36}$$

- *If $\eta_t \leq C/t < 2/\beta$, then*

$$\sum_{t=1}^{T}\eta_t\prod_{j=t+1}^{T}(1 + \beta\eta_j)^2 \leq Ce^{2C\beta}T^{2C\beta}\min\left\{1 + \frac{1}{2C\beta}, \log(eT)\right\}. \tag{37}$$

**Proof.**

- If $\eta_t = C \leq 2/(\beta + \gamma)$ then

$$\sum_{t=1}^{T} \eta_t \prod_{j=t+1}^{T} (1 - \eta_j \gamma) = C \sum_{t=1}^{T} (1 - C\gamma)^{T-t} = C(1 - C\gamma)^T \sum_{t=1}^{T} (1 - C\gamma)^{-t}$$

$$= C \frac{1 - (1 - C\gamma)^T}{C\gamma} = \frac{1 - (1 - C\gamma)^T}{\gamma},$$

- If $\eta_t = C/t \leq 2/(\beta + \gamma)$, for some $C \geq 2/\gamma$ for $t \geq 1 + \lceil \frac{\beta}{\gamma} \rceil$ and $\eta_t = C'/t \leq 2/(\beta + \gamma)$ for some $C' < 2/(\gamma + \beta)$ for $t \leq \lceil \frac{\beta}{\gamma} \rceil$ then

$$\sum_{t=1}^{T} \eta_t \prod_{j=t+1}^{T} \left(1 - \frac{\eta_j \gamma}{2}\right)$$

$$\leq \sum_{t=1}^{\lceil \frac{\beta}{\gamma} \rceil} \frac{C'}{t} \prod_{j=t+1}^{T} \left(1 - \frac{C'\gamma}{2j}\right) + \sum_{t=1+\lceil \frac{\beta}{\gamma} \rceil}^{T} \frac{C}{t} \prod_{j=t+1}^{T} \left(1 - \frac{1}{j}\right)$$

$$= \sum_{t=1}^{\lceil \frac{\beta}{\gamma} \rceil} \frac{C'}{t} \prod_{j=t+1}^{T} \left(1 - \frac{C'\gamma}{2j}\right) + \sum_{t=1+\lceil \frac{\beta}{\gamma} \rceil}^{T} \frac{C}{t} \frac{t}{T} \leq \sum_{t=1}^{\lceil \frac{\beta}{\gamma} \rceil} \frac{C'}{t} + C \frac{\left[T - \lceil \frac{\beta}{\gamma} \rceil\right]_+}{T}$$

$$\leq C \left(1 + \log(\lceil \beta/\gamma \rceil) + \left[1 - \lceil \frac{\beta}{T\gamma} \rceil\right]_+\right) \leq C \log \left(e^2 \lceil \beta/\gamma \rceil\right).$$

- If $\eta_t \leq C/t \leq 2/\beta$, then

$$\sum_{t=1}^{T} \eta_t \prod_{j=t+1}^{T} (1 + \beta \eta_j)^2 = \sum_{t=1}^{T} \frac{C}{t} \prod_{j=t+1}^{T} \left(1 + \beta \frac{C}{j}\right)^2$$

$$\leq \sum_{t=1}^{T} \frac{C}{t} \prod_{j=t+1}^{T} \exp\left(2\beta \frac{C}{j}\right)$$

$$= \sum_{t=1}^{T} \frac{C}{t} \exp\left(2\beta \sum_{j=t+1}^{T} \frac{C}{j}\right)$$

$$\leq \sum_{t=1}^{T} \frac{C}{t} \exp\left(2C\beta \left(\log(T) + 1 - \log(t+1)\right)\right)$$

$$= Ce^{2C\beta} T^{2C\beta} \sum_{t=1}^{T} \frac{1}{t} \frac{1}{(t+1)^{2C\beta}}$$

$$\leq Ce^{2C\beta} T^{2C\beta} \sum_{t=1}^{T} \frac{1}{t} \frac{1}{(t+1)^{2C\beta}}$$

$$\leq Ce^{2C\beta} T^{2C\beta} \sum_{t=1}^{T} \frac{1}{t^{1+2C\beta}} \qquad (38)$$

$$= Ce^{2C\beta} T^{2C\beta} \left(1 + \sum_{t=2}^{T} \frac{1}{t^{1+2C\beta}}\right)$$

$$\leq Ce^{2C\beta} T^{2C\beta} \left(1 + \int_1^T \frac{1}{x^{1+2C\beta}} dx\right)$$

$$= Ce^{2C\beta}T^{2C\beta}\left(1 + \frac{1}{2C\beta}\left(1 - T^{-2C\beta}\right)\right)$$

$$= Ce^{2C\beta}T^{2C\beta}\left(1 + \frac{1}{2C\beta}\right) - C\frac{e^{2C\beta}}{2\beta}$$

$$\leq Ce^{2C\beta}T^{2C\beta}\left(1 + \frac{1}{2C\beta}\right), \tag{39}$$

additionally $\sum_{t=1}^{T} 1/t \leq \log(eT)$, thus the term in the inequality 38 may be upper bounded by $Ce^{2C\beta}T^{2C\beta}\log(eT)$ for any $T \in \mathbb{N}$, and we conclude that

$$\sum_{t=1}^{T} \eta_t \prod_{j=t+1}^{T} (1 + \beta\eta_j)^2 \leq Ce^{2C\beta}T^{2C\beta} \min\left\{1 + \frac{1}{2C\beta}, \log(eT)\right\}. \tag{40}$$

The last inequality completes the proof. $\qquad\square$

In the next section we prove the stability and generalization error bounds for nonconvex losses.

## C  NONCONVEX LOSS: PROOF OF THEOREM 6 & THEOREM 7

Let $z_1, z_2, \ldots, z_i, \ldots, z_n, z_i'$ be i.i.d. random variables, define $S \triangleq (z_1, z_2, \ldots, z_i, \ldots, z_n)$ and $S^{(i)} \triangleq (z_1, z_2, \ldots, z_i', \ldots, z_n)$, $W_1 = W_1'$. The updates for any $t \geq 1$ are

$$W_{t+1} = W_t - \frac{\eta_t}{n}\sum_{j=1}^{n}\nabla f(W_t, z_j), \tag{41}$$

$$W_{t+1}^{(i)} = W_t^{(i)} - \frac{\eta_t}{n}\sum_{j=1, j\neq i}^{n}\nabla f(W_t^{(i)}, z_j) - \frac{\eta_t}{n}\nabla f(W_t^{(i)}, z_i'). \tag{42}$$

Then for any $t \geq 1$, we derive the stability recursion as

$$\|W_{t+1} - W_{t+1}^{(i)}\|_2$$

$$\leq \|W_t - W_t^{(i)}\|_2 + \frac{\eta_t}{n}\left\|\sum_{j=1, j\neq i}^{n}\left(\nabla f(W_t, z_j) - \nabla f(W_t^{(i)}, z_j)\right)\right\|_2$$

$$\quad + \frac{\eta_t}{n}\|\nabla f(W_t, z_i) - \nabla f(W_t^{(i)}, z_i')\|_2$$

$$\leq \|W_t - W_t^{(i)}\|_2 + \frac{\eta_t}{n}\sum_{j=1, j\neq i}^{n}\|\nabla f(W_t, z_j) - \nabla f(W_t^{(i)}, z_j)\|_2$$

$$\quad + \frac{\eta_t}{n}\left(\|\nabla f(W_t, z_i)\|_2 + \|\nabla f(W_t^{(i)}, z_i')\|_2\right)$$

$$\leq \|W_t - W_t^{(i)}\|_2 + \frac{\eta_t(n-1)}{n}\beta\|W_t - W_t^{(i)}\|_2 + \frac{\eta_t}{n}\left(\|\nabla f(W_t, z_i)\|_2 + \|\nabla f(W_t^{(i)}, z_i')\|_2\right) \tag{43}$$

$$= \left(1 + \frac{n-1}{n}\beta\eta_t\right)\|W_t - W_t^{(i)}\|_2 + \frac{\eta_t}{n}\left(\|\nabla f(W_t, z_i)\|_2 + \|\nabla f(W_t^{(i)}, z_i')\|_2\right), \tag{44}$$

inequality 43 comes from the smoothness of the loss. Then by solving the recursion we find

$$\|W_{T+1} - W_{T+1}^{(i)}\|_2$$

$$\leq \frac{1}{n}\sum_{t=1}^{T}\eta_t\left(\|\nabla f(W_t, z_i)\|_2 + \|\nabla f(W_t^{(i)}, z_i')\|_2\right)\prod_{j=t+1}^{T}\left(1 + \frac{n-1}{n}\beta\eta_j\right)$$

$$\leq \frac{1}{n}\sum_{t=1}^{T}\eta_t\left(\|\nabla f(W_t, z_i)\|_2 + \|\nabla f(W_t^{(i)}, z_i')\|_2\right)\prod_{j=t+1}^{T}(1 + \beta\eta_j)$$

$$\leq \frac{1}{n} \sqrt{\sum_{t=1}^{T} \eta_t \left( \|\nabla f(W_t, z_i)\|_2 + \|\nabla f(W_t^{(i)}, z_i')\|_2 \right)^2 \sum_{t=1}^{T} \eta_t \prod_{j=t+1}^{T} (1 + \beta \eta_j)^2}$$

$$\leq \frac{\sqrt{2}}{n} \sqrt{\sum_{t=1}^{T} \eta_t \left( \|\nabla f(W_t, z_i)\|_2^2 + \|\nabla f(W_t^{(i)}, z_i')\|_2^2 \right) \sum_{t=1}^{T} \eta_t \prod_{j=t+1}^{T} (1 + \beta \eta_j)^2}.$$

The last display gives

$$\|W_{T+1} - W_{T+1}^{(i)}\|_2^2 \leq \frac{2}{n^2} \sum_{t=1}^{T} \eta_t \left( \|\nabla f(W_t, z_i)\|_2^2 + \|\nabla f(W_t^{(i)}, z_i')\|_2^2 \right) \sum_{t=1}^{T} \eta_t \prod_{j=t+1}^{T} (1 + \beta \eta_j)^2 ,$$

and by taking the expectation we find

$$\mathbb{E}[\|W_{T+1} - W_{T+1}^{(i)}\|_2^2]$$

$$\leq \frac{2}{n^2} \sum_{t=1}^{T} \eta_t \left( \mathbb{E}[\|\nabla f(W_t, z_i)\|_2^2] + \mathbb{E}[\|\nabla f(W_t^{(i)}, z_i')\|_2^2] \right) \sum_{t=1}^{T} \eta_t \prod_{j=t+1}^{T} (1 + \beta \eta_j)^2$$

$$\leq \frac{4 \epsilon_{\text{path}}}{n^2} \sum_{t=1}^{T} \eta_t \prod_{j=t+1}^{T} (1 + \beta \eta_j)^2 . \tag{45}$$

We evaluate the summation of the products in the inequality 45. Lemma 15 under the choice of decreasing learning rate $\eta_t \leq C/t \leq 2/\beta$ shows that

$$\sum_{t=1}^{T} \eta_t \prod_{j=t+1}^{T} (1 + \beta \eta_j)^2 \leq C e^{2C\beta} T^{2C\beta} \min \left\{ 1 + \frac{1}{2C\beta}, \log(eT) \right\}. \tag{46}$$

Through the inequalities 45, 46 and Theorem 3, we derive the bound on the generalization error as

$$|\epsilon_{\text{gen}}|$$

$$\leq 2 \sqrt{2\beta(\epsilon_{\text{opt}} + \epsilon_{\mathbf{c}})\epsilon_{\text{stab}(A)}} + 2\beta \epsilon_{\text{stab}(A)}$$

$$\leq \frac{4}{n} \sqrt{2\beta(\epsilon_{\text{opt}} + \epsilon_{\mathbf{c}})\epsilon_{\text{path}} \sum_{t=1}^{T} \eta_t \prod_{j=t+1}^{T} (1 + \beta \eta_j)^2 + 8\beta \frac{\epsilon_{\text{path}}}{n^2} \sum_{t=1}^{T} \eta_t \prod_{j=t+1}^{T} (1 + \beta \eta_j)^2}$$

$$\leq \frac{4}{n} \sqrt{2C\beta(\epsilon_{\text{opt}} + \epsilon_{\mathbf{c}})\epsilon_{\text{path}} e^{C\beta} T^{C\beta} \min \left\{ 1 + \frac{1}{2C\beta}, \log(eT) \right\}^{\frac{1}{2}}}$$

$$+ 8C\beta \frac{\epsilon_{\text{path}}}{n^2} e^{2C\beta} T^{2C\beta} \min \left\{ 1 + \frac{1}{2C\beta}, \log(eT) \right\}$$

Under the choice $\eta_t \leq C/t < 1/\beta$ for all $t$, we choose $C < 1/\beta$, further we define $\epsilon \triangleq \beta C < 1$, and $\bar{C}(\epsilon, T) \triangleq \min \{ \epsilon + 1/2, \epsilon \log(eT) \}$ to get

$$|\epsilon_{\text{gen}}| \leq \frac{4\sqrt{2}}{n} \sqrt{(\epsilon_{\text{opt}} + \epsilon_{\mathbf{c}})\epsilon_{\text{path}} (eT)^{\epsilon} \bar{C}^{\frac{1}{2}}(\epsilon, T) + 8 \frac{\epsilon_{\text{path}}}{n^2} (eT)^{2\epsilon} \bar{C}(\epsilon, T)}$$

$$\leq \frac{4\sqrt{3}}{n} \sqrt{(\epsilon_{\text{opt}} + \epsilon_{\mathbf{c}})\epsilon_{\text{path}} (eT)^{\epsilon} + 12 \frac{\epsilon_{\text{path}}}{n^2} (eT)^{2\epsilon}}. \tag{47}$$

The last inequality provide the generalization error bound and completes the proof. $\qquad \square$

Next we derive upper bounds on expected path error $\epsilon_{\text{path}}$ and optimization error $\epsilon_{\text{opt}}$, to show an alternative expression of the generalization error inequality 47. We continue by proving the proof of Corollary 8.

## C.1 PROOF OF COROLLARY 8.

The self-bounding property of the non-negative $\beta$-smooth loss function $f(\cdot; z)$ (Srebro et al., 2010, Lemma 3.1) gives $\|\nabla f(W_t, z_i)\|_2^2 \leq 4\beta f(W_t, z_i)$. By taking expectation, and through the Assumption 1 and the equation 20 we find

$$\mathbb{E}[\|\nabla f(W_t, z_i)\|_2^2] \leq 4\beta \mathbb{E}[f(W_t, z_i)] = 4\beta \mathbb{E}[R_S(W_t)]. \tag{48}$$

The definition of $\epsilon_{\text{path}}$ (Definition 5), and the decreasing learning rate ($\eta_t = C/t < 1/\beta t$) give

$$\epsilon_{\text{path}} \triangleq \sum_{t=1}^{T} \eta_t \mathbb{E}[\|\nabla f(W_t, z_i)\|_2^2] \leq 4\beta \sum_{t=1}^{T} \eta_t \mathbb{E}[R_S(W_t)]$$

$$\leq 4\beta \mathbb{E}[R_S(W_1)] \sum_{t=1}^{T} \eta_t \tag{49}$$

$$< 4\mathbb{E}[R_S(W_1)] \sum_{t=1}^{T} \frac{1}{t}$$

$$\leq 4\mathbb{E}[R_S(W_1)] \log(eT), \tag{50}$$

and the inequality 49 holds since the learning rate $\eta_t < 2/\beta$ guarantees descent at each iteration. Similarly, $\epsilon_{\text{opt}} + \epsilon_{\mathbf{c}} \leq \mathbb{E}[R_S(W_1)]$. The last inequality together with the inequalities 50 and 47 give

$$|\epsilon_{\text{gen}}| \leq \frac{4\sqrt{3}}{n} \sqrt{(\epsilon_{\text{opt}} + \epsilon_{\mathbf{c}})\epsilon_{\text{path}}} (eT)^{\epsilon} + 12 \frac{\epsilon_{\text{path}}}{n^2} (eT)^{2\epsilon}$$

$$\leq \left( \frac{8\sqrt{3}}{n} \sqrt{\log(eT)} (eT)^{\epsilon} + \frac{48}{n^2} \log(eT)(eT)^{2\epsilon} \right) \mathbb{E}[R_S(W_1)].$$

The last inequality provides the bound of the corollary.

## D PL OBJECTIVE

Herein we provide the proofs of the results associated with the PL condition on the objective. We start by proving an upper bound on the average output stability. Then by combining Lemma 16 and Theorem 3 we derive generalization error bounds for symmetric algorithms and smooth losses, as well as the generalization error bound of the full-batch GD under the PL condition. A similar proof technique of the next lemma also appears in prior work by Lei et al. (Lei & Ying, 2020a, Proof of Lemma B.2).

**Lemma 16** *Let the loss function $f(\cdot; z)$ be non-negative, nonconvex and $\beta$-smooth for all $z \in \mathcal{Z}$. Further, let the objective be $\mu$-PL, $\mathbb{E}[\|\nabla R_S(w)\|_2^2] \geq 2\mu \mathbb{E}[R_S(w) - R_S^*]$ for all $w \in \mathbb{R}^d$. Then for any algorithm it is true that*

$$\mathbb{E}[\|A(S^{(i)}) - A(S)\|_2^2] \leq \frac{16}{\mu} \epsilon_{\text{opt}} + \frac{8\beta}{n^2 \mu^2} \left( \mathbb{E}[R_S(\pi_S)] + \mathbb{E}[R(\pi_S)] \right). \tag{51}$$

**Proof.** Define the projection $\pi_{S^{(i)}} \triangleq \pi(A(S^{(i)}))$ of the point $A(S^{(i)})$ to the set of the minimizers of $R_{S^{(i)}}(\cdot)$, and the similarly the projection $\pi_S \triangleq \pi(A(S))$ of the point $A(S)$ to the set of the minimizers of $R_S(\cdot)$. Then

$$\mathbb{E}[\|A(S^{(i)}) - A(S)\|_2^2]$$

$$\leq 4\mathbb{E}[\|A(S^{(i)}) - \pi_{S^{(i)}}\|_2^2] + 4\mathbb{E}[\|A(S) - \pi_S\|_2^2] + 2\mathbb{E}[\|\pi_{S^{(i)}} - \pi_S\|_2^2]$$

$$\leq \frac{8}{\mu} \mathbb{E}[R_{S^{(i)}}(A(S^{(i)})) - R_{S^{(i)}}^*] + \frac{8}{\mu} \mathbb{E}[R_S(A(S)) - R_S^*] + 2\mathbb{E}[\|\pi_{S^{(i)}} - \pi_S\|_2^2] \tag{52}$$

$$= \frac{16}{\mu} \epsilon_{\text{opt}} + 2\mathbb{E}[\|\pi_{S^{(i)}} - \pi_S\|_2^2]$$

$$\leq \frac{16}{\mu} \epsilon_{\text{opt}} + \frac{4}{\mu} \left( \mathbb{E}[R_S(\pi_{S^{(i)}})] - \mathbb{E}[R_S(\pi_S)] \right), \tag{53}$$

the inequalities 52 and 53 come from the quadratic growth (Karimi et al., 2016). Recall that, the PL condition on the objective gives

$$\frac{1}{2\mu}\mathbb{E}[\|\nabla R_S(\pi_{S^{(i)}})\|_2^2] \geq \mathbb{E}[R_S(\pi_{S^{(i)}}) - R_S(\pi_S)]. \tag{54}$$

We combine the inequalities 53 and 54 to find

$$\mathbb{E}[\|A(S^{(i)}) - A(S)\|_2^2] \leq \frac{16}{\mu}\epsilon_{\text{opt}} + \frac{2}{\mu^2}\mathbb{E}[\|\nabla R_S(\pi_{S^{(i)}})\|_2^2]. \tag{55}$$

Also, it is true that

$$\|\nabla R_S(\pi_{S^{(i)}})\|_2^2 = \|\nabla R_{S^{(i)}}(\pi_{S^{(i)}}) - \frac{1}{n}\nabla f(\pi_{S^{(i)}}; z_i') + \frac{1}{n}\nabla f(\pi_{S^{(i)}}; z_i)\|_2^2$$

$$= \frac{2}{n^2}\|\nabla f(\pi_{S^{(i)}}; z_i')\|_2^2 + \frac{2}{n^2}\|\nabla f(\pi_{S^{(i)}}; z_i)\|_2^2 \tag{56}$$

$$\leq \frac{4\beta}{n^2}f(\pi_{S^{(i)}}; z_i') + \frac{4\beta}{n^2}f(\pi_{S^{(i)}}; z_i), \tag{57}$$

equation 56 holds because $\nabla R_{S^{(i)}}(\pi_{S^{(i)}}) = 0$, the inequality 57 holds for nonnegative losses (Srebro et al., 2010, (Lemma 3.1). Through inequality57 we find,

$$\mathbb{E}[\|\nabla R_S(\pi_{S^{(i)}})\|_2^2] \leq \frac{4\beta}{n^2}\mathbb{E}[f(\pi_{S^{(i)}}; z_i')] + \frac{4\beta}{n^2}\mathbb{E}[f(\pi_{S^{(i)}}; z_i)] \tag{58}$$

$$= \frac{4\beta}{n^2}\mathbb{E}[f(\pi_S; z_i)] + \frac{4\beta}{n^2}\mathbb{E}[f(\pi_S; z_i')], \tag{59}$$

and the last equality holds because $z_i, z_i'$ are exchangeable. We combine the inequalities 55 and 59 to find

$$\frac{1}{n}\sum_{i=1}^{n}\mathbb{E}[\|A(S^{(i)}) - A(S)\|_2^2] \leq \frac{16}{\mu}\epsilon_{\text{opt}} + \frac{8\beta}{n^2\mu^2}\left(\frac{1}{n}\sum_{i=1}^{n}\mathbb{E}[f(\pi_S; z_i)] + \frac{1}{n}\sum_{i=1}^{n}\mathbb{E}[f(\pi_S; z_i')]\right) \tag{60}$$

$$= \frac{16}{\mu}\epsilon_{\text{opt}} + \frac{8\beta}{n^2\mu^2}\left(\mathbb{E}[R_S(\pi_S)] + \mathbb{E}[R(\pi_S)]\right). \tag{61}$$

Since $\mathbb{E}[\|A(S^{(i)}) - A(S)\|_2^2] = \mathbb{E}[\|A(S^{(j)}) - A(S)\|_2^2]$ for any $i, j \in \{1, \ldots, n\}$, we conclude that for any $i \in \{1, \ldots, n\}$

$$\mathbb{E}[\|A(S^{(i)}) - A(S)\|_2^2] \leq \frac{16}{\mu}\epsilon_{\text{opt}} + \frac{8\beta}{n^2\mu^2}\left(\mathbb{E}[R_S(\pi_S)] + \mathbb{E}[R(\pi_S)]\right). \tag{62}$$

The last inequality provides the bound on the expected stability and completes the proof. □

**Corollary 17** *Let $\pi_S \triangleq \pi(A(S))$ be the projection of the point $A(S)$ to the set of the minimizers of $R_S(\cdot)$. Further, define the constant $\tilde{c} \triangleq \mathbb{E}[R_S(\pi_S) + R(\pi_S)]$. For any symmetric algorithm, non-negative $\beta$-smooth loss function $f(\cdot; z)$ for all $z \in \mathcal{Z}$, $\mu$-PL objective and $\mathbb{E}[R_S^*] = 0$, it is true that*

$$|\epsilon_{\text{gen}}| \leq \frac{8\beta\sqrt{\tilde{c}}}{n\mu}\sqrt{\epsilon_{\text{opt}}} + \frac{16\beta^2}{n^2\mu^2}\tilde{c} + \frac{44\beta}{\mu}\epsilon_{\text{opt}}. \tag{63}$$

*Further, define the constant $c \triangleq 44\max\{\mathbb{E}[R_S(\pi_S) + R(\pi_S)], \mathbb{E}[R_S(W_1) - R_S^*]\}$. Then the generalization error of the full-batch GD with step-size choice $\eta_t = 1/\beta$ and $T$ total number of iterations is bounded as follows*

$$|\epsilon_{\text{gen}}| \leq \frac{c\beta}{\mu}\frac{\left(1 - \frac{\mu}{\beta}\right)^{T/2}}{n} + \frac{c\beta^2}{n^2\mu^2} + \frac{c\beta}{\mu}\left(1 - \frac{\mu}{\beta}\right)^T. \tag{64}$$

**Proof.** We define the constant $\tilde{c} \triangleq \mathbb{E}[R_S(\pi_S) + R(\pi_S)]$ apply Theorem 3 and Lemma 16 to find

$$
\begin{aligned}
&|\epsilon_{\text{gen}}| \\
&\leq 2\sqrt{2\beta(\epsilon_{\text{opt}} + \epsilon_{\mathbf{c}})\epsilon_{\text{stab}(A)}} + 2\beta\epsilon_{\text{stab}(A)} \\
&\leq \left(\frac{8}{\sqrt{\mu}}\sqrt{\epsilon_{\text{opt}}} + \frac{4\sqrt{2\beta\tilde{c}}}{n\mu}\right)\sqrt{2\beta(\epsilon_{\text{opt}} + \epsilon_{\mathbf{c}})} + \frac{32\beta}{\mu}\epsilon_{\text{opt}} + \frac{16\beta^2}{n^2\mu^2}\tilde{c} \\
&\leq \frac{8\sqrt{2\beta\epsilon_{\text{opt}}}}{\sqrt{\mu}}\sqrt{\epsilon_{\text{opt}} + \epsilon_{\mathbf{c}}} + \frac{8\beta\sqrt{\tilde{c}}}{n\mu}\sqrt{\epsilon_{\text{opt}} + \epsilon_{\mathbf{c}}} + \frac{32\beta}{\mu}\epsilon_{\text{opt}} + \frac{16\beta^2}{n^2\mu^2}\tilde{c} \\
&\leq \frac{8\beta\sqrt{\tilde{c}}}{n\mu}\sqrt{\epsilon_{\text{opt}} + \epsilon_{\mathbf{c}}} + \frac{8\sqrt{2\beta\epsilon_{\text{opt}}\epsilon_{\mathbf{c}}}}{\sqrt{\mu}} + \frac{16\beta^2}{n^2\mu^2}\tilde{c} + \frac{44\beta}{\mu}\epsilon_{\text{opt}}.
\end{aligned}
\tag{65}
$$

The last inequality completes the proof. $\qquad \square$

## E CONVEX LOSS: PROOF OF THEOREM 9 AND THEOREM 10.

We start by proving the non-expansive property of the stability iterates for the case of $\beta$-smooth convex loss. Then we continue with the proof of the stability generalization error.

**Lemma 18** *Let the gradient of the loss be $\beta$-Lipschitz for all $z \in \mathcal{Z}$. If the loss function is convex and $\eta_t < 2/\beta$, then for any $t \leq T + 1$ the updates $W_t, W_t^{(i)}$ satisfy the next inequality*

$$
\left\| W_t - W_t^{(i)} - \frac{\eta_t}{n}\sum_{j=1, j\neq i}^{n}\left(\nabla f(W_t, z_j) - \nabla f(W_t^{(i)}, z_j)\right)\right\|_2^2 \leq \|W_t - W_t^{(i)}\|_2^2.
\tag{66}
$$

**Proof.** By the definition of $\beta$-Lipschitz gradients and triangle inequality, it is true that

$$
\|\nabla f(W_t, z_j) - \nabla f(W_t^{(i)}, z_j)\|_2 \leq \beta\|W_t - W_t^{(i)}\|_2 \implies
\tag{67}
$$

$$
\|\sum_{j\in\mathcal{J}}\nabla f(W_t, z_j) - \sum_{j\in\mathcal{J}}\nabla f(W_t^{(i)}, z_j)\|_2 \leq \beta|\mathcal{J}|\|W_t - W_t^{(i)}\|_2.
\tag{68}
$$

Since the function $h(W) \triangleq \sum_{j\in\mathcal{J}}\nabla f(W, z_j)$ is convex and the gradient of $h(w)$ is $\beta|\mathcal{J}|$-Lipschitz, it follows that (co-coersivity of the gradient)

$$
\sum_{j\in\mathcal{J}}\langle\nabla f(W_t, z_j) - \nabla f(W_t^{(i)}, z_j), W_t - W_t^{(i)}\rangle
\tag{69}
$$

$$
\geq \frac{1}{\beta|\mathcal{J}|}\|\sum_{j\in\mathcal{J}}\nabla f(W_t, z_j) - \sum_{j\in\mathcal{J}}\nabla f(W_t^{(i)}, z_j)\|_2^2.
\tag{70}
$$

Then prove the inequality 66 as follows

$$
\begin{aligned}
&\left\| W_t - W_t^{(i)} - \frac{\eta_t}{n}\sum_{j=1, j\neq i}^{n}\left(\nabla f(W_t, z_j) - \nabla f(W_t^{(i)}, z_j)\right)\right\|_2^2 \\
&= \|W_t - W_t^{(i)}\|_2^2 - 2\frac{\eta_t}{n}\sum_{j=1, j\neq i}^{n}\langle\nabla f(W_t, z_j) - \nabla f(W_t^{(i)}, z_j), W_t - W_t^{(i)}\rangle \\
&\quad + \frac{\eta_t^2}{n^2}\|\sum_{j=1, j\neq i}^{n}\left(\nabla f(W_t, z_j) - \nabla f(W_t^{(i)}, z_j)\right)\|_2^2 \\
&= \|W_t - W_t^{(i)}\|_2^2 - 2\frac{\eta_t}{n}\langle\sum_{j=1, j\neq i}^{n}\nabla f(W_t, z_j) - \sum_{j=1, j\neq i}^{n}\nabla f(W_t^{(i)}, z_j), W_t - W_t^{(i)}\rangle \\
&\quad + \frac{\eta_t^2}{n^2}\|\sum_{j=1, j\neq i}^{n}\nabla f(W_t, z_j) - \sum_{j=1, j\neq i}^{n}\nabla f(W_t^{(i)}, z_j)\|_2^2
\end{aligned}
\tag{71}
$$

$$\leq \|W_t - W_t^{(i)}\|_2^2 - 2\frac{\eta_t}{\beta(n-1)n}\|\sum_{j=1,j\neq i}^{n} \nabla f(W_t, z_j) - \sum_{j=1,j\neq i}^{n} \nabla f(W_t^{(i)}, z_j)\|_2^2$$

$$+ \frac{\eta_t^2}{n^2}\|\sum_{j=1,j\neq i}^{n} \nabla f(W_t, z_j) - \sum_{j=1,j\neq i}^{n} \nabla f(W_t^{(i)}, z_j)\|_2^2 \qquad (72)$$

$$= \|W_t - W_t^{(i)}\|_2^2 + \frac{\eta_t}{n}\left(\frac{\eta_t}{n} - \frac{2}{\beta(n-1)}\right)\left\|\sum_{j=1,j\neq i}^{n}\left(\nabla f(W_t, z_j) - \nabla f(W_t^{(i)}, z_j)\right)\right\|_2^2$$

$$\leq \|W_t - W_t^{(i)}\|_2^2, \qquad (73)$$

equation 71 holds from the expansion of the squared norm, 72 comes from the inequality 70. The inequality 73 holds under the choice $\eta_t < 2/\beta$ and completes the proof. $\square$

**Lemma 19 (Accumulated Path Error - Convex Loss)** *Let the loss function $f(\cdot; z)$ be convex and $\beta$-smooth and $\eta_t \leq 1/2\beta$. Then the expected path-error of the full-batch GD after $T$ iterations is bounded as*

$$\epsilon_{\text{path}} \leq 4\beta\mathbb{E}[\|W_1 - W_S^*\|_2^2] + 8\beta\mathbb{E}[R_S(W_S^*)]\sum_{t=1}^{T}\eta_t \qquad (74)$$

**Proof.** The self-bounding property of the non-negative $\beta$-smooth loss function $f(\cdot; z)$ (Srebro et al., 2010, Lemma 3.1) gives $\|\nabla f(W_t, z_i)\|_2^2 \leq 4\beta f(W_t, z_i)$. By taking expectation, and through the equation 20 we find

$$\mathbb{E}[\|\nabla f(W_t, z_i)\|_2^2] \leq 4\beta\mathbb{E}[f(W_t, z_i)] = 4\beta\mathbb{E}[R_S(W_t)]. \qquad (75)$$

Similarly to the approach by Lei & Ying (2020b, Appendix A, Lemma 2), we use the convexity and the assumption $\eta_t \leq 1/2\beta$ to find

$$\begin{aligned}
\|W_{t+1} - W_S^*\|_2^2 &= \|W_t - \eta_t\nabla R_S(W_t) - W_S^*\|_2^2 \\
&= \|W_t - W_S^*\|_2^2 + \eta_t^2\|\nabla R_S(W_t)\|_2^2 + 2\eta_t\langle W_S^* - W_t, \nabla R_S(W_t)\rangle \\
&\leq \|W_t - W_S^*\|_2^2 + \eta_t^2\|\nabla R_S(W_t)\|_2^2 + 2\eta_t\left(R_S(W_S^*) - R_S(W_t)\right) \\
&\leq \|W_t - W_S^*\|_2^2 + 2\beta\eta_t^2 R_S(W_t) + 2\eta_t\left(R_S(W_S^*) - R_S(W_t)\right) \\
&\leq \|W_t - W_S^*\|_2^2 + 2\eta_t R_S(W_S^*) - \eta_t R_S(W_t).
\end{aligned}$$

The last gives

$$\sum_{t=1}^{T}\eta_t R_S(W_t) \leq \sum_{t=1}^{T}\|W_t - W_S^*\|_2^2 - \sum_{t=1}^{T}\|W_{t+1} - W_S^*\|_2^2 + 2\sum_{t=1}^{T}\eta_t R_S(W_S^*)$$

$$\leq \|W_1 - W_S^*\|_2^2 + 2\sum_{t=1}^{T}\eta_t R_S(W_S^*). \qquad (76)$$

The definition of $\epsilon_{\text{path}}$ (Definition 5), the inequalities 75, 76 and the choice of the learning rate ($\eta_t \leq 1/2\beta$) give

$$\epsilon_{\text{path}} \triangleq \sum_{t=1}^{T}\eta_t\mathbb{E}[\|\nabla f(W_t, z_i)\|_2^2] \leq 4\beta\sum_{t=1}^{T}\eta_t\mathbb{E}[R_S(W_t)]$$

$$\leq 4\beta\mathbb{E}[\|W_1 - W_S^*\|_2^2] + 8\beta\sum_{t=1}^{T}\eta_t\mathbb{E}[R_S(W_S^*)]. \qquad (77)$$

The last inequality provides the bound on the $\epsilon_{\text{path}}$. $\square$

The standard choice of $\eta_t \leq 1/\beta$ gives the next known bound on the optimization error.

**Lemma 20 (Optimization Error - Convex Loss (Nesterov, 1998))** *If $f(\cdot; z)$ is a convex and $\beta$-smooth function and $\eta_t \leq 1/\beta$, then*

$$\epsilon_{\text{opt}} = \mathbb{E}[R_S(A(S)) - R_S(W_S^*)] \leq \frac{\mathbb{E}[\|W_1 - W_S^*\|_2^2]}{\sum_{t=1}^{T}\eta_t\left(1 - \frac{\beta\eta_t}{2}\right)}. \qquad (78)$$

### E.1 Proof of Theorem 9 and Theorem 10

Let $z_1, z_2, \ldots, z_i, \ldots, z_n, z_i'$ be i.i.d. random variables, define $S \triangleq (z_1, z_2, \ldots, z_i, \ldots, z_n)$ and $S^{(i)} \triangleq (z_1, z_2, \ldots, z_i', \ldots, z_n)$, $W_1 = W_1'$. The updates for any $t \geq 1$ are

$$W_{t+1} = W_t - \frac{\eta_t}{n} \sum_{j=1}^{n} \nabla f(W_t, z_j), \tag{79}$$

$$W_{t+1}^{(i)} = W_t^{(i)} - \frac{\eta_t}{n} \sum_{j=1, j \neq i}^{n} \nabla f(W_t^{(i)}, z_j) - \frac{\eta_t}{n} \nabla f(W_t^{(i)}, z_i'). \tag{80}$$

Then for any $t \geq 1$

$$\begin{aligned}
&\|W_{t+1} - W_{t+1}^{(i)}\|_2 \\
&\leq \left\| W_t - W_t^{(i)} - \frac{\eta_t}{n} \sum_{j=1, j \neq i}^{n} \left( \nabla f(W_t, z_j) - \nabla f(W_t^{(i)}, z_j) \right) \right\|_2 \\
&\quad + \frac{\eta_t}{n} \|\nabla f(W_t, z_i) - \nabla f(W_t^{(i)}, z_i')\|_2 \\
&\leq \sqrt{\left\| W_t - W_t^{(i)} - \frac{\eta_t}{n} \sum_{j=1, j \neq i}^{n} \left( \nabla f(W_t, z_j) - \nabla f(W_t^{(i)}, z_j) \right) \right\|_2^2} \\
&\quad + \frac{\eta_t}{n} \left( \|\nabla f(W_t, z_i)\|_2 + \|\nabla f(W_t^{(i)}, z_i')\|_2 \right) \\
&\leq \|W_t - W_t^{(i)}\|_2 + \frac{\eta_t}{n} \left( \|\nabla f(W_t, z_i)\|_2 + \|\nabla f(W_t^{(i)}, z_i')\|_2 \right). \tag{81}
\end{aligned}$$

The inequality 81 comes from Lemma 18. Then by solving the recursion, we find

$$\|W_{T+1} - W_{T+1}^{(i)}\|_2 \leq \frac{1}{n} \sum_{t=1}^{T} \eta_t \left( \|\nabla f(W_t, z_i)\|_2 + \|\nabla f(W_t^{(i)}, z_i')\|_2 \right)$$

thus

$$\begin{aligned}
\|W_{T+1} - W_{T+1}^{(i)}\|_2^2 &\leq \frac{1}{n^2} \left( \sum_{t=1}^{T} \eta_t \left( \|\nabla f(W_t, z_i)\|_2 + \|\nabla f(W_t^{(i)}, z_i')\|_2 \right) \right)^2 \\
&\leq \frac{2}{n^2} \sum_{t=1}^{T} \eta_t \left( \|\nabla f(W_t, z_i)\|_2^2 + \|\nabla f(W_t^{(i)}, z_i')\|_2^2 \right) \sum_{t=1}^{T} \eta_t. \tag{82}
\end{aligned}$$

Inequality 82 gives that for any $i \in \{1, \ldots, n\}$

$$\begin{aligned}
\mathbb{E}[\|W_{T+1} - W_{T+1}^{(i)}\|_2^2] &\leq \frac{2}{n^2} \sum_{t=1}^{T} \eta_t \left( \mathbb{E}[\|\nabla f(W_t, z_i)\|_2^2] + \mathbb{E}[\|\nabla f(W_t^{(i)}, z_i')\|_2^2] \right) \sum_{t=1}^{T} \eta_t \\
&= \frac{4}{n^2} \sum_{t=1}^{T} \eta_t \mathbb{E}[\|\nabla f(W_t, z_i)\|_2^2] \sum_{t=1}^{T} \eta_t = \frac{4\epsilon_{\text{path}}}{n^2} \sum_{t=1}^{T} \eta_t. \tag{83}
\end{aligned}$$

Recall that $W_{T+1} \equiv A(S)$ and $W_{T+1}^{(i)} \equiv A(S^{(i)})$. Theorem 3 and the inequality 83 give

$$\begin{aligned}
|\epsilon_{\text{gen}}| &\leq 2\sqrt{2\beta \left( \epsilon_{\text{opt}} + \mathbb{E}[R_S(W_S^*)] \right) \epsilon_{\text{stab}(A)}} + 2\beta \epsilon_{\text{stab}(A)} \\
&\leq 2\sqrt{2\beta \left( \epsilon_{\text{opt}} + \mathbb{E}[R_S(W_S^*)] \right) \frac{4\epsilon_{\text{path}}}{n^2} \sum_{t=1}^{T} \eta_t} + 2\beta \frac{4\epsilon_{\text{path}}}{n^2} \sum_{t=1}^{T} \eta_t \\
&= \frac{4\sqrt{\left( \epsilon_{\text{opt}} + \mathbb{E}[R_S(W_S^*)] \right) \epsilon_{\text{path}}}}{n} \sqrt{2\beta \sum_{t=1}^{T} \eta_t + 8\beta \frac{\epsilon_{\text{path}}}{n^2} \sum_{t=1}^{T} \eta_t}. \tag{84}
\end{aligned}$$

Under the choice of constant learning rate $\eta_t = 1/2\beta$, Lemma 19 together with the inequality 83 give $\epsilon_{\text{path}} \leq 4\beta\mathbb{E}[\|W_1 - W_S^*\|_2^2] + 8\beta\mathbb{E}[R_S(W_S^*)] \sum_{t=1}^{T} \eta_t$, and $\epsilon_{\text{opt}} \leq 3\beta\mathbb{E}[\|W_1 - W_S^*\|_2^2]/T$. Thus

$$\epsilon_{\text{stab}(A)} \leq \frac{32}{n^2} \left( \frac{\beta}{2}\mathbb{E}[\|W_1 - W_S^*\|_2^2] + \beta\mathbb{E}[R_S(W_S^*)] \sum_{t=1}^{T} \eta_t \right) \sum_{t=1}^{T} \eta_t \tag{85}$$

$$= \frac{32}{n^2} \left( \frac{\beta}{2}\mathbb{E}[\|W_1 - W_S^*\|_2^2] + \frac{1}{2}\mathbb{E}[R_S(W_S^*)]T \right) \frac{T}{2\beta} \tag{86}$$

$$= \frac{8T}{n^2} \left( \mathbb{E}[\|W_1 - W_S^*\|_2^2] + \mathbb{E}[R_S(W_S^*)]\frac{T}{\beta} \right). \tag{87}$$

The inequality 84 and Lemma 20 give

$$|\epsilon_{\text{gen}}| \leq 2\sqrt{2\beta \left( \epsilon_{\text{opt}} + \mathbb{E}[R_S(W_S^*)] \right) \epsilon_{\text{stab}(A)}} + 2\beta\epsilon_{\text{stab}(A)}$$

$$\leq 2\sqrt{2\beta \left( \epsilon_{\text{opt}} + \mathbb{E}[R_S(W_S^*)] \right) \frac{8T}{n^2} \left( \mathbb{E}[\|W_1 - W_S^*\|_2^2] + \mathbb{E}[R_S(W_S^*)]\frac{T}{\beta} \right)}$$

$$+ 2\beta\frac{8T}{n^2} \left( \mathbb{E}[\|W_1 - W_S^*\|_2^2] + \mathbb{E}[R_S(W_S^*)]\frac{T}{\beta} \right)$$

$$\leq \frac{8\sqrt{T}}{n}\sqrt{\left( \epsilon_{\text{opt}} + \mathbb{E}[R_S(W_S^*)] \right) \left( \beta\mathbb{E}[\|W_1 - W_S^*\|_2^2] + T\mathbb{E}[R_S(W_S^*)] \right)}$$

$$+ \frac{16T}{n^2} \left( \beta\mathbb{E}[\|W_1 - W_S^*\|_2^2] + T\mathbb{E}[R_S(W_S^*)] \right)$$

$$\leq \frac{8\sqrt{T}}{n}\sqrt{\left( \frac{3\beta\mathbb{E}[\|W_1 - W_S^*\|_2^2]}{T} + \mathbb{E}[R_S(W_S^*)] \right) \left( \beta\mathbb{E}[\|W_1 - W_S^*\|_2^2] + T\mathbb{E}[R_S(W_S^*)] \right)}$$

$$+ \frac{16T}{n^2} \left( \beta\mathbb{E}[\|W_1 - W_S^*\|_2^2] + T\mathbb{E}[R_S(W_S^*)] \right)$$

$$= \frac{8}{n}\sqrt{\left( 3\beta\mathbb{E}[\|W_1 - W_S^*\|_2^2] + T\mathbb{E}[R_S(W_S^*)] \right) \left( \beta\mathbb{E}[\|W_1 - W_S^*\|_2^2] + T\mathbb{E}[R_S(W_S^*)] \right)}$$

$$+ \frac{16T}{n^2} \left( \beta\mathbb{E}[\|W_1 - W_S^*\|_2^2] + T\mathbb{E}[R_S(W_S^*)] \right)$$

$$\leq \frac{8}{n} \left( 3\beta\mathbb{E}[\|W_1 - W_S^*\|_2^2] + T\mathbb{E}[R_S(W_S^*)] \right) + \frac{16T}{n^2} \left( \beta\mathbb{E}[\|W_1 - W_S^*\|_2^2] + T\mathbb{E}[R_S(W_S^*)] \right)$$

$$\leq 8 \left( \frac{1}{n} + \frac{2T}{n^2} \right) \left( 3\beta\mathbb{E}[\|W_1 - W_S^*\|_2^2] + T\mathbb{E}[R_S(W_S^*)] \right).$$

The last inequality completes the proof. $\qquad\square$

## F   STRONGLY-CONVEX OBJECTIVE: PROOF OF THEOREM 12 AND THEOREM 13

Similarly to the convex case, first we provide the contractive property of the stability recursion in the strongly convex loss case. Then we prove the stability and generalization error bounds.

**Lemma 21** *Let the objective function be $\gamma$-strongly convex ($\gamma > 0$) and the leave-one-out objective function be $\gamma_{loo}$-strongly convex for some $\gamma_{loo} \geq 0$. If the loss function is convex $\beta$-smooth for all $z \in \mathcal{Z}$ and $\eta_t \leq 2/(\beta + \gamma)$, then for any $t \leq T + 1$ the updates $W_t, W_t^{(i)}$ satisfy the inequality*

$$\left\| W_t - W_t^{(i)} - \eta_t \left( \nabla R_{S^{-i}}(W_t) - \nabla R_{S^{-i}}(W_t^{(i)}) \right) \right\|_2^2 \leq (1 - \eta_t\gamma_{loo}) \|W_t - W_t^{(i)}\|_2^2.$$

**Proof.**   The function $R_{S^{-i}}(\cdot)$ is also $\beta$-smooth for all $z \in \mathcal{Z}$ and the strong convexity gives

$$\langle \nabla R_{S^{-i}}(W_t) - \nabla R_{S^{-i}}(W_t^{(i)}), W_t - W_t^{(i)} \rangle$$

$$\geq \frac{\beta \gamma_{loo}}{\beta + \gamma_{loo}} \|W_t - W_t^{(i)}\|_2^2 + \frac{1}{(\beta + \gamma_{loo})} \|\nabla R_{S^{-i}}(W_t) - \nabla R_{S^{-i}}(W_t^{(i)})\|_2^2 \tag{88}$$

We expand the squared norm as follows

$$\left\| W_t - W_t^{(i)} - \eta_t \left( \nabla R_{S^{-i}}(W_t) - \nabla R_{S^{-i}}(W_t^{(i)}) \right) \right\|_2^2$$

$$= \|W_t - W_t^{(i)}\|_2^2 - 2\eta_t \langle \nabla R_{S^{-i}}(W_t) - \nabla R_{S^{-i}}(W_t^{(i)}), W_t - W_t^{(i)} \rangle$$
$$+ \eta_t^2 \|\nabla R_{S^{-i}}(W_t) - \nabla R_{S^{-i}}(W_t^{(i)})\|_2^2$$

$$\leq \|W_t - W_t^{(i)}\|_2^2 + \eta_t^2 \|\nabla R_{S^{-i}}(W_t) - \nabla R_{S^{-i}}(W_t^{(i)})\|_2^2$$
$$- 2\eta_t \left( \frac{\beta \gamma_{loo}}{\beta + \gamma_{loo}} \|W_t - W_t^{(i)}\|_2^2 + \frac{1}{(\beta + \gamma_{loo})} \|\nabla R_{S^{-i}}(W_t) - \nabla R_{S^{-i}}(W_t^{(i)})\|_2^2 \right) \tag{89}$$

$$= \left( 1 - 2\eta_t \frac{\beta \gamma_{loo}}{\beta + \gamma_{loo}} \right) \|W_t - W_t^{(i)}\|_2^2$$
$$+ \eta_t \left( \eta_t - \frac{2}{\beta + \gamma_{loo}} \right) \|\nabla R_{S^{-i}}(W_t) - \nabla R_{S^{-i}}(W_t^{(i)})\|_2^2$$

$$\leq \left( 1 - 2\eta_t \frac{\beta \gamma_{loo}}{\beta + \gamma_{loo}} \right) \|W_t - W_t^{(i)}\|_2^2. \tag{90}$$

We apply the inequality 88 to derive 89. The inequality 90 holds since $\eta_t \leq 2/(\beta + \gamma)$ and $\beta \geq \gamma > \gamma_{loo}$. Also

$$2\eta_t \frac{\beta \gamma_{loo}}{\beta + \gamma_{loo}} \geq 2\eta_t \frac{\beta \gamma_{loo}}{2\beta} = \eta_t \gamma_{loo}. \tag{91}$$

Through the inequalities 90 and 91 to derive the bound of the lemma. $\qquad \square$

**Lemma 22 (Accumulated Path Error - Strongly Convex Loss)** *Let the objective function $R_S(\cdot)$ be $\gamma$-strongly convex and $\beta$-smooth. Define $\Gamma(\gamma, T) \triangleq (1 - \exp(\frac{-4T\gamma}{\beta+\gamma}))/(\exp(\frac{-4\gamma}{\beta+\gamma}) - 1)$. If $\eta_t = 2/(\beta + \gamma)$, then the expected path-error of the full-batch GD after $T$ iterations are bounded as*

$$\epsilon_{\text{path}} \leq \frac{4\beta^2}{\beta + \gamma} \Gamma(\gamma, T) \mathbb{E}[\|W_1 - W_S^*\|_2^2] + \frac{8\beta T}{\beta + \gamma} \mathbb{E}[R_S(W_S^*)], \tag{92}$$

**Proof.** The self-bounding property of the non-negative $\beta$-smooth loss function $f(\cdot; z)$ (Srebro et al., 2010, Lemma 3.1) gives $\|\nabla f(W_t, z_i)\|_2^2 \leq 4\beta f(W_t, z_i)$. By taking expectation, and through the Assumption 1 and equation 20, we find

$$\mathbb{E}[\|\nabla f(W_t, z_i)\|_2^2] \leq 4\beta \mathbb{E}[f(W_t, z_i)] = 4\beta \mathbb{E}[R_S(W_t)] = 4\beta \mathbb{E}[R_S(W_t) - R_S(W_S^*) + R_S(W_S^*)]. \tag{93}$$

Further, Lemma 23 and the choice of constant learning rate $\eta = 2/(\beta + \gamma)$ give

$$\mathbb{E}[R_S(W_t) - R_S(W_S^*)] \leq \frac{\beta}{2} \exp\left( \frac{-4t}{\frac{\beta}{\gamma} + 1} \right) \mathbb{E}[\|W_1 - W_S^*\|_2^2]. \tag{94}$$

The definition of $\epsilon_{\text{path}}$ (Definition 5), the inequalities 93 and 94 and the constant learning rate ($\eta_t = 2/(\beta + \gamma)$) give

$$\epsilon_{\text{path}} \triangleq \sum_{t=1}^{T} \eta_t \mathbb{E}[\|\nabla f(W_t, z_i)\|_2^2] \tag{95}$$

$$\leq 4\beta \sum_{t=1}^{T} \eta_t \mathbb{E}[R_S(W_t) - R_S(W_S^*) + R_S(W_S^*)]$$

$$\leq 4\beta \sum_{t=1}^{T} \frac{2}{\beta + \gamma} \frac{\beta}{2} \exp\left( \frac{-4t}{\frac{\beta}{\gamma} + 1} \right) \mathbb{E}[\|W_1 - W_S^*\|_2^2] + \frac{8\beta T}{\beta + \gamma} \mathbb{E}[R_S(W_S^*)]$$

$$\leq \frac{4\beta^2}{\beta+\gamma}\mathbb{E}[\|W_1 - W_S^*\|_2^2]\sum_{t=1}^{T}\exp\left(\frac{-4t}{\frac{\beta}{\gamma}+1}\right) + \frac{8\beta T}{\beta+\gamma}\mathbb{E}[R_S(W_S^*)]$$

$$= \frac{4\beta^2}{\beta+\gamma}\mathbb{E}[\|W_1 - W_S^*\|_2^2]\underbrace{\exp\left(\frac{-4}{\frac{\beta}{\gamma}+1}\right)\frac{1 - \exp\left(\frac{-4T}{\frac{\beta}{\gamma}+1}\right)}{1 - \exp\left(\frac{-4}{\frac{\beta}{\gamma}+1}\right)}}_{\Gamma(\gamma,T)} + \frac{8\beta T}{\beta+\gamma}\mathbb{E}[R_S(W_S^*)]$$

$$= \frac{4\beta^2}{\beta+\gamma}\Gamma(\gamma,T)\mathbb{E}[\|W_1 - W_S^*\|_2^2] + \frac{8\beta T}{\beta+\gamma}\mathbb{E}[R_S(W_S^*)]. \tag{96}$$

The last inequality provides the bound on the $\epsilon_{\text{path}}$. Further, we can show that

$$\Gamma(\gamma,T) \leq \min\left\{\frac{1}{e^{\frac{4\gamma}{\beta+\gamma}} - 1}, T\right\} \tag{97}$$

to simplify the expression in the inequality 96. $\square$

**Lemma 23 ((Nesterov, 1998, Theorem 2.1.14))** *If $f(\cdot; z)$ is a $\gamma$-strongly convex and $\beta$-smooth function and $\eta_t = 2/(\beta + \gamma)$, then*

$$\epsilon_{\text{opt}} \leq \frac{\beta}{2}\exp\left(\frac{-4T}{\frac{\beta}{\gamma}+1}\right)\mathbb{E}[\|W_1 - W_S^*\|_2^2]. \tag{98}$$

*Alternatively, if $\eta_t = c/t$, then*

$$\epsilon_{\text{opt}} \leq \frac{\beta}{2}T^{-\frac{2c\beta\gamma}{\beta+\gamma}}\mathbb{E}[\|W_1 - W_S^*\|_2^2]. \tag{99}$$

### F.1 PROOF OF THEOREM 12 AND THEOREM 13

Let $z_1, z_2, \ldots, z_i, \ldots, z_n, z_i'$ be i.i.d. random variables, define $S \triangleq (z_1, z_2, \ldots, z_i, \ldots, z_n)$ and $S^{(i)} \triangleq (z_1, z_2, \ldots, z_i', \ldots, z_n)$, $W_1 = W_1'$. The updates for any $t \geq 1$ are

$$W_{t+1} = W_t - \frac{\eta_t}{n}\sum_{j=1}^{n}\nabla f(W_t, z_j), \tag{100}$$

$$W_{t+1}^{(i)} = W_t^{(i)} - \frac{\eta_t}{n}\sum_{j=1, j\neq i}^{n}\nabla f(W_t^{(i)}, z_j) - \frac{\eta_t}{n}\nabla f(W_t^{(i)}, z_i'). \tag{101}$$

Then similarly to the inequality 81 we get

$$\|W_{t+1} - W_{t+1}^{(i)}\|_2$$
$$\leq \left\|W_t - W_t^{(i)} - \frac{\eta_t}{n}\sum_{j=1, j\neq i}^{n}\left(\nabla f(W_t, z_j) - \nabla f(W_t^{(i)}, z_j)\right)\right\|_2$$
$$\quad + \frac{\eta_t}{n}\|\nabla f(W_t, z_i) - \nabla f(W_t^{(i)}, z_i')\|_2$$
$$\leq \sqrt{\left\|W_t - W_t^{(i)} - \eta_t\left(R_{S^{-i}}(W_t) - R_{S^{-i}}(W_t^{(i)})\right)\right\|_2^2}$$
$$\quad + \frac{\eta_t}{n}\left(\|\nabla f(W_t, z_i)\|_2 + \|\nabla f(W_t^{(i)}, z_i')\|_2\right)$$
$$\leq (1 - \eta_t\gamma_{loo})^{\frac{1}{2}}\|W_t - W_t^{(i)}\|_2 + \frac{\eta_t}{n}\left(\|\nabla f(W_t, z_i)\|_2 + \|\nabla f(W_t^{(i)}, z_i')\|_2\right) \tag{102}$$

and we apply Lemma 21 to derive the bound in 102. Then by solving the recursion we find

$$\|W_{T+1} - W_{T+1}^{(i)}\|_2$$

$$\leq \frac{1}{n} \sum_{t=1}^{T} \eta_t \left( \|\nabla f(W_t, z_i)\|_2 + \|\nabla f(W_t^{(i)}, z_i')\|_2 \right) \prod_{j=t+1}^{T} (1 - \eta_j \gamma_{loo})^{\frac{1}{2}}$$

$$\leq \frac{1}{n} \sqrt{\sum_{t=1}^{T} \eta_t \left( \|\nabla f(W_t, z_i)\|_2 + \|\nabla f(W_t^{(i)}, z_i')\|_2 \right)^2 \sum_{t=1}^{T} \eta_t \prod_{j=t+1}^{T} (1 - \eta_j \gamma_{loo})}$$

$$\leq \frac{1}{n} \sqrt{2 \sum_{t=1}^{T} \eta_t \left( \|\nabla f(W_t, z_i)\|_2^2 + \|\nabla f(W_t^{(i)}, z_i')\|_2^2 \right) \sum_{t=1}^{T} \eta_t \prod_{j=t+1}^{T} (1 - \eta_j \gamma_{loo})}.$$

The last inequality provides the stability bound

$$\|W_{T+1} - W_{T+1}^{(i)}\|_2^2$$
$$\leq \frac{2}{n^2} \sum_{t=1}^{T} \eta_t \left( \|\nabla f(W_t, z_i)\|_2^2 + \|\nabla f(W_t^{(i)}, z_i')\|_2^2 \right) \sum_{t=1}^{T} \eta_t \prod_{j=t+1}^{T} (1 - \eta_j \gamma_{loo}). \tag{103}$$

Inequality 103 gives that for any $i \in \{1, \dots, n\}$

$$\mathbb{E}[\|W_{T+1} - W_{T+1}^{(i)}\|_2^2]$$
$$\leq \frac{2}{n^2} \sum_{t=1}^{T} \eta_t \left( \mathbb{E}[\|\nabla f(W_t, z_i)\|_2^2] + \mathbb{E}[\|\nabla f(W_t^{(i)}, z_i')\|_2^2] \right) \sum_{t=1}^{T} \eta_t \prod_{j=t+1}^{T} (1 - \eta_j \gamma_{loo})$$
$$= \frac{4}{n^2} \sum_{t=1}^{T} \eta_t \mathbb{E}[\|\nabla f(W_t, z_i)\|_2^2] \sum_{t=1}^{T} \eta_t \prod_{j=t+1}^{T} (1 - \eta_j \gamma_{loo})$$
$$= \frac{4\epsilon_{\text{path}}}{n^2} \sum_{t=1}^{T} \eta_t \prod_{j=t+1}^{T} (1 - \eta_j \gamma_{loo}). \tag{104}$$

Recall that $W_{T+1} \equiv A(S)$ and $W_{T+1}^{(i)} \equiv A(S^{(i)})$. Due to space limitation, we define $\Omega(\eta_t, \gamma_{loo}) \triangleq \sum_{t=1}^{T} \eta_t \prod_{j=t+1}^{T} \left(1 - \eta_t \gamma_{\text{loo}}^2 / \beta \right)$ Theorem 3 and the inequality 104 give

$$|\epsilon_{\text{gen}}| \leq 2\sqrt{2\beta(\epsilon_{\text{opt}} + \mathbb{E}[R_S(W_S^*)])\epsilon_{\text{stab}(A)}} + 2\beta\epsilon_{\text{stab}(A)} \tag{105}$$

$$\leq 2\sqrt{2\beta(\epsilon_{\text{opt}} + \mathbb{E}[R_S(W_S^*)])\frac{4\epsilon_{\text{path}}}{n^2}\Omega(\eta_t, \gamma_{loo})} + 2\beta\frac{4\epsilon_{\text{path}}}{n^2}\Omega(\eta_t, \gamma_{loo})$$

$$= \frac{4\sqrt{(\epsilon_{\text{opt}} + \mathbb{E}[R_S(W_S^*)])\epsilon_{\text{path}}}}{n}\sqrt{2\beta\Omega(\eta_t, \gamma_{loo})} + 8\beta\frac{\epsilon_{\text{path}}}{n^2}\Omega(\eta_t, \gamma_{loo}).$$

Under the choice of $\eta_t = C = \frac{2}{\beta+\gamma} < \frac{2}{\beta}$, the inequality 104, Lemmata 15 and 22 and give

$$\mathbb{E}[\|A(S) - A(S^{(i)})\|_2^2] \leq \frac{4\epsilon_{\text{path}}}{n^2} \sum_{t=1}^{T} \eta_t \prod_{j=t+1}^{T} (1 - \eta_t \gamma_{\text{loo}})$$

$$= \frac{4\epsilon_{\text{path}}}{n^2} \underbrace{\frac{1 - \left(1 - \frac{2\gamma_{loo}}{\beta+\gamma}\right)^T}{\gamma_{loo}}}_{\Lambda(\gamma_{loo}, T)}$$

$$\leq \frac{4\epsilon_{\text{path}}}{n^2} \Lambda(\gamma_{loo}, T) \leq \frac{4\epsilon_{\text{path}}}{n^2} \min\left\{\frac{1}{\gamma_{loo}}, \frac{2T}{\beta}\right\} \tag{106}$$

and the last inequality holds since $\Lambda(\gamma_{loo}, T) \leq 1/\gamma_{loo}$ for any $T$ and the monotonicity of $\Lambda(\gamma_{loo}, T)$ gives $\Lambda(\gamma_{loo}, T) \leq 2T/\beta$ for any pair $\gamma_{loo} \leq \gamma$. Though the inequality 105, we find the generalization error bound

$$|\epsilon_{\text{gen}}| \leq \frac{4\sqrt{(\epsilon_{\text{opt}} + \mathbb{E}[R_S(W_S^*)])\epsilon_{\text{path}}}}{n}\sqrt{2\beta\Omega(\eta_t, \gamma_{loo})} + 8\beta\frac{\epsilon_{\text{path}}}{n^2}\Omega(\eta_t, \gamma_{loo})$$

$$\leq \frac{4\sqrt{(\epsilon_{\text{opt}} + \mathbb{E}[R_S(W_S^*)])\epsilon_{\text{path}}}}{n} \sqrt{2\beta\Lambda(\gamma_{loo}, T)} + 8\beta \frac{\epsilon_{\text{path}}}{n^2} \Lambda(\gamma_{loo}, T)$$

$$\leq \frac{4}{n} \sqrt{\left(\frac{\beta}{2} \exp\left(\frac{-4T}{\frac{\beta}{\gamma} + 1}\right) \mathbb{E}[\|W_1 - W_S^*\|_2^2] + \mathbb{E}[R_S(W_S^*)]\right)}$$

$$\times \sqrt{\left(\frac{4\beta^2}{\beta + \gamma}\Gamma(\gamma, T)\mathbb{E}[\|W_1 - W_S^*\|_2^2] + \frac{8\beta T}{\beta + \gamma}\mathbb{E}[R_S(W_S^*)]\right)} \sqrt{2\beta\Lambda(\gamma_{loo}, T)}$$

$$+ 8\beta \frac{\frac{4\beta^2}{\beta+\gamma}\Gamma(\gamma, T)\mathbb{E}[\|W_1 - W_S^*\|_2^2] + \frac{8\beta T}{\beta+\gamma}\mathbb{E}[R_S(W_S^*)]}{n^2} \Lambda(\gamma_{loo}, T)$$

$$\leq \frac{4}{n} \sqrt{\left(\frac{4\beta}{\beta + \gamma}\Gamma(\gamma, T) + \frac{8\beta T}{\beta + \gamma}\right) \max\left\{\beta\mathbb{E}[\|W_1 - W_S^*\|_2^2], \mathbb{E}[R_S(W_S^*)]\right\}}$$

$$\times \sqrt{2\beta\Lambda(\gamma_{loo}, T)} + \left(\frac{4}{n}\sqrt{\frac{1}{2}\exp\left(\frac{-4T\gamma}{\beta + \gamma}\right)\left(\frac{4\beta}{\beta + \gamma}\Gamma(\gamma, T) + \frac{8\beta T}{\beta + \gamma}\right)}\sqrt{2\beta\Lambda(\gamma_{loo}, T)}\right.$$

$$\left. + 8\frac{\frac{4\beta}{\beta+\gamma}\Gamma(\gamma, T)] + \frac{8\beta T}{\beta+\gamma}}{n^2}\beta\Lambda(\gamma_{loo}, T)\right) \max\{\beta\mathbb{E}[\|W_1 - W_S^*\|_2^2], \mathbb{E}[R_S(W_S^*)]\}$$

$$= \frac{4}{n} \sqrt{\left(\frac{4\beta}{\beta + \gamma}\Gamma(\gamma, T) + \frac{8\beta T}{\beta + \gamma}\right) \max\left\{\beta\mathbb{E}[\|W_1 - W_S^*\|_2^2], \mathbb{E}[R_S(W_S^*)]\right\}}$$

$$\times \sqrt{2\beta\Lambda(\gamma_{loo}, T)} + \left(\frac{4}{n}\sqrt{\exp\left(\frac{-4T}{\frac{\beta}{\gamma} + 1}\right)\left(\frac{4\beta}{\beta + \gamma}\Gamma(\gamma, T) + \frac{8\beta T}{\beta + \gamma}\right)}\sqrt{\beta\Lambda(\gamma_{loo}, T)}\right.$$

$$\left. + 8\frac{\frac{4\beta}{\beta+\gamma}\Gamma(\gamma, T) + \frac{8\beta T}{\beta+\gamma}}{n^2}\beta\Lambda(\gamma_{loo}, T)\right) \max\{\beta\mathbb{E}[\|W_1 - W_S^*\|_2^2], \mathbb{E}[R_S(W_S^*)]\}. \tag{107}$$

We proceed by applying the upper bounds of $\Gamma(\gamma, T), \Lambda(\gamma_{loo}, T)$ as appear in the inequalities 97 and 106 respectively and equation 107 gives

$$|\epsilon_{\text{gen}}|$$

$$\leq \frac{4}{n} \sqrt{\left(\frac{4\beta}{\beta + \gamma} \min\left\{\frac{1}{e^{\frac{4\gamma}{\beta+\gamma}} - 1}, T\right\} + \frac{8\beta T}{\beta + \gamma}\right)}$$

$$\times \sqrt{\max\left\{\beta\mathbb{E}[\|W_1 - W_S^*\|_2^2], \mathbb{E}[R_S(W_S^*)]\right\} 2 \min\left\{\frac{\beta}{\gamma_{loo}}, 2T\right\}}$$

$$+ \left(\frac{4}{n}\exp\left(\frac{-2T\gamma}{\beta + \gamma}\right)\sqrt{\left(\frac{4\beta}{\beta + \gamma} \min\left\{\frac{1}{e^{\frac{4\gamma}{\beta+\gamma}} - 1}, T\right\} + \frac{8\beta T}{\beta + \gamma}\right) 2 \min\left\{\frac{\beta}{\gamma_{loo}}, 2T\right\}}\right.$$

$$\left. + 8\frac{\frac{4\beta}{\beta+\gamma}\min\left\{\frac{1}{e^{\frac{4\gamma}{\beta+\gamma}}-1}, T\right\} + \frac{8\beta T}{\beta+\gamma}}{n^2} \min\left\{\frac{\beta}{\gamma_{loo}}, 2T\right\}\right) \max\{\beta\mathbb{E}[\|W_1 - W_S^*\|_2^2], \mathbb{E}[R_S(W_S^*)]\}$$

$$\leq \frac{4}{n} \sqrt{\left(\frac{4\beta T}{\beta + \gamma} + \frac{8\beta T}{\beta + \gamma}\right) \max\left\{\beta\mathbb{E}[\|W_1 - W_S^*\|_2^2], \mathbb{E}[R_S(W_S^*)]\right\} 2 \min\left\{\frac{\beta}{\gamma_{loo}}, 2T\right\}}$$

$$+ \left(\frac{4}{n}\exp\left(\frac{-2T\gamma}{\beta + \gamma}\right)\sqrt{\left(\frac{4\beta T}{\beta + \gamma} + \frac{8\beta T}{\beta + \gamma}\right) 2 \min\left\{\frac{\beta}{\gamma_{loo}}, 2T\right\}}\right.$$

$$\left. + 8\frac{\frac{4\beta T}{\beta+\gamma} + \frac{8\beta T}{\beta+\gamma}}{n^2} \min\left\{\frac{\beta}{\gamma_{loo}}, 2T\right\}\right) \max\{\beta\mathbb{E}[\|W_1 - W_S^*\|_2^2], \mathbb{E}[R_S(W_S^*)]\}$$

$$
= \frac{8\sqrt{3}}{n} \sqrt{\frac{\beta T}{\beta + \gamma} \max\left\{\beta \mathbb{E}[\|W_1 - W_S^*\|_2^2], \mathbb{E}[R_S(W_S^*)]\right\} 2 \min\left\{\frac{\beta}{\gamma_{loo}}, 2T\right\}}
$$

$$
+ \left(\frac{8\sqrt{3}}{n} \exp\left(\frac{-2T\gamma}{\beta + \gamma}\right) \sqrt{\frac{\beta T}{\beta + \gamma} 2 \min\left\{\frac{\beta}{\gamma_{loo}}, 2T\right\}}\right.
$$

$$
\left. + 96 \frac{\frac{\beta T}{\beta + \gamma}}{n^2} \min\left\{\frac{\beta}{\gamma_{loo}}, 2T\right\}\right) \max\{\beta \mathbb{E}[\|W_1 - W_S^*\|_2^2], \mathbb{E}[R_S(W_S^*)]\}. \tag{108}
$$

To simplify the last display, we define the terms $\mathrm{m}(\gamma_{loo}, T) \triangleq \beta T \min\{\beta/\gamma_{loo}, 2T\}/(\beta + \gamma)$ and $\mathrm{M}(W_1) \triangleq \max\left\{\beta \mathbb{E}[\|W_1 - W_S^*\|_2^2], \mathbb{E}[R_S(W_S^*)]\right\}$

$$
|\epsilon_{\mathrm{gen}}|
$$

$$
\leq \frac{8\sqrt{6}}{n} \sqrt{\mathrm{m}(\gamma_{loo}, T) M(W_1)}
$$

$$
+ \left(\frac{8\sqrt{6}}{n} \exp\left(\frac{-2T\gamma}{\beta + \gamma}\right) \sqrt{\mathrm{m}(\gamma_{loo}, T)} + \frac{96}{n^2} \mathrm{m}(\gamma_{loo}, T)\right) \mathrm{M}(W_1)
$$

$$
= \frac{8\sqrt{6}}{n} \left(\sqrt{M(W_1)} + \left(\exp\left(\frac{-2T\gamma}{\beta + \gamma}\right) + \frac{4\sqrt{3}}{n} \sqrt{\mathrm{m}(\gamma_{loo}, T)}\right) \mathrm{M}(W_1)\right) \sqrt{\mathrm{m}(\gamma_{loo}, T)}
$$

Choose $T = \log(n)(\beta + \gamma)/2\gamma$ and define $\mathrm{m}_{n,\gamma_{loo}} \triangleq \frac{\beta}{2\gamma} \min\left\{\frac{\beta}{\gamma_{loo}}, \frac{\beta + \gamma}{\gamma} \log n\right\}$, then the inequality 108 gives

$$
|\epsilon_{\mathrm{gen}}|
$$

$$
\leq \frac{8\sqrt{6\log n}}{n} \sqrt{\frac{\beta}{2\gamma} \max\left\{\beta \mathbb{E}[\|W_1 - W_S^*\|_2^2], \mathbb{E}[R_S(W_S^*)]\right\} \min\left\{\frac{\beta}{\gamma_{loo}}, \frac{\beta + \gamma}{\gamma} \log n\right\}}
$$

$$
+ \left(\frac{8\sqrt{6}}{n^2} \sqrt{\frac{\beta \log n}{2\gamma} \min\left\{\frac{\beta}{\gamma_{loo}}, \frac{\beta + \gamma}{\gamma} \log(n)\right\}}\right.
$$

$$
\left. + \frac{48\beta}{\gamma} \frac{\log n}{n^2} \min\left\{\frac{\beta}{\gamma_{loo}}, \frac{\beta + \gamma}{\gamma} \log(n)\right\}\right) \max\{\beta \mathbb{E}[\|W_1 - W_S^*\|_2^2], \mathbb{E}[R_S(W_S^*)]\}
$$

$$
= \frac{8\sqrt{6\log n}}{n} \left(\sqrt{\mathrm{m}_{n,\gamma_{loo}} \mathrm{M}(W_1)} + \left(\frac{1}{n} \sqrt{\mathrm{m}_{n,\gamma_{loo}}} + \frac{4\sqrt{3}}{n} \mathrm{m}_{n,\gamma_{loo}}\right) \mathrm{M}(W_1)\right)
$$

$$
= \frac{8\sqrt{6\log n}}{n} \left(\sqrt{\mathrm{M}(W_1)} + \frac{1 + 4\sqrt{3\mathrm{m}_{n,\gamma_{loo}}}}{n} \mathrm{M}(W_1)\right) \sqrt{\mathrm{m}_{n,\gamma_{loo}}}. \tag{109}
$$

The last inequality completes the proof. $\qquad\square$

