# OpenReview forum: "Beyond Lipschitz: Sharp Generalization and Excess Risk Bounds for Full-Batch GD"
_ICLR.cc/2023/Conference — ICLR 2023 poster_

### Official Review · Reviewer_Phd7 · 2022-10-20

**Confidence:** 3
**Correctness:** 3
**Technical Novelty And Significance:** 3
**Empirical Novelty And Significance:** Not applicable
**Recommendation:** 8

**Clarity, Quality, Novelty And Reproducibility:**

* **Clarity**: The paper is clear and well-written.

* **Quality**: The quality of the paper is good. The ideas are interesting and well-executed.

* **Novelty**: The first theorem (Theorem 3) is not novel, but the proof is. Then, the theory for full-batch GD (the main part of the paper) is novel.

* **Reproducibility**: \
*Theory*: I reproduced all the proofs except for the questions that I placed the authors in the weaknesses. \
*Experiments*: There are no experiments.

**Strength And Weaknesses:**

**Strengths**

* The paper is generally well written and easy to follow. It feels that this would be true even for people that are only somewhat familiar with these kind of results.
* The results for full-batch GD in all of the settings are stronger than previously known in the literature. They show that GD can converge faster or equally fast and to better or equal rates than SGD for smooth and both convex and strongly convex losses (which is in line with current experimental results).
* The appearance of the path error in their proofs and the way it is bounded is interesting and provides insights on what makes full-batch GD perform as it does.

**Weaknesses**

* As mentioned in the text after Theorem 3, (4) is essentially the tightest version of Theorem 2 b) of *[Lei and Ying 2020b]*. However, it would be nice to clarify that this is only up to constants, since the first term in (4) would be divided by 2 and the second by 4 if one were to obtain the variant in that paper.
* In the remark about the particularization of Theorem 3 under the Polyak-Łojasiewicz (PL) condition, it would be nice to write what PL stands for and to cite the original references from *[Polyak 1963]* and *[Łojasiewicz 1963]* or the paper that coined the term *[Karimi et al. 2016]*. Also, the way the discussion after (5) is written, although correct, is a little confusing. Could you please re-write or explain further the arguments under the interpolation regime and $\beta \leq n \mu / 4$?
* After Corollary 8 it is said that (9) matches bounds in prior work, including information-theoretic bounds for the SGLD algorithm and it cites *[Wang et al. 2021b]*. I could not see to which bound the text refers after reading the reference, could you point that out and, in case that it needs some further modification, explain that?
* Throughout the paper it is said that the only assumptions are smoothness (and after convexity and strong convexity) of the loss, and that the algorithm is symmetric and deterministic. Then, apparently dimension-independent complexity of the generalization and the risk are provided. However, the bounds depend on the standard term $\mathbb{E}[\lVert W_1 - W_S^\star \rVert_2^2]$, which usually depends on the dimension and is only in $\mathcal{O}(1)$ when the parameter space is bounded by some constant.\
An important element for the bounds of gradient methods under the Lipschitz/smooth assumptions is that they are dimension-independent when the parameter space is bounded. This is also the case here and I believe it should be mentioned clearly before showing the complexity terms that can be derived from these bounds.

* There are some claims that I did not understood. Please, could you explain/justify them to me/in the text a little more?

  * Why does (51) hold? I understand that you use the quadratic growth property in $\lVert A(S) - \pi_S \rVert_2^2$, but I don't see why that would hold for $\lVert \pi_{S^{(i)}} - \pi_S \rVert_2^2$. Both $\pi_S$ and $\pi_{S^{(i)}}$ are projections, of $A(S)$ and $A(S^{(i)})$, but not the projections of each other.
  * Why does (89) hold?
  * In (101) you use $1- \frac{\eta_j \gamma}{2}$ and then you replace that in the next array of equations for $(1-\eta_j \gamma_{loo})$. Why can you do that?
  * Why is true that $\Lambda(\gamma_{loo}, T) \leq 2 T / \beta$?
  * How do you obtain the third inequality of page 29? I am particularly surprised by the $\mathbb{E}[R_S(W_S^\star)]$ before the max inside the squared root.

**References**

*[Polyak 1963] Gradient methods for minimizing functionals (in Russian).* \
*[Łojasiewicz 1963] A topological property of real analytic subsets (in French).* \
*[Karimi et al. 2016] Linear convergence of gradient and proximal-
gradient methods under the Polyak- Łojasiewicz condition.* \
*[Lei and Ying 2020b] Fine-grained analysis of stability and generalization for stochastic gradient descent.* \
*[Wang et al. 2021b] Analyzing the generalization capability of SGLD using properties of gaussian channels.*


**Summary Of The Paper:**

This provides us with expected generalization and excess risk guarantees for symmetric, deterministic algorithms on smooth loss functions. These bounds depend on the $\ell_2$ expected output stability, the expected optimization error, and the model capacity (hence, Theorem 3 and its proof are interesting contributions).

The bounds are specialized to full-batch gradient descent (GD) in the case of non-convex, convex, and strongly convex functions. Interestingly, the resulting bounds from this paper show that:

* When the loss is convex, GD can reach an expected risk in $\mathcal{O}(1/\sqrt{n})$ in $T=\sqrt{n}$ iterations, as opposed to SGD that requires $T=n$ iterations, albeit with a different choice of the learning rate.
* When the loss is convex, GD can also reach an expected risk in $\mathcal{O}(1/n)$ in $T=n$ iterations like SGD with the same learning rate.
* When the loss is strongly convex, GD can reach a risk of $\mathcal{O}(\sqrt{\log n}/n)$ in only $\Theta(\log n)$ iterations, while SGD requires $\Theta(n)$ iterations to reach a risk in $\mathcal{O}(1/n)$.

At the core of the proofs for the bounds on full-batch GD is the path error, which is defined as the cumulative sum of the squared norm of the gradient over iterations, weighted by the learning rate. This is also an important contribution, that is in line with other works from the literature in the idea that the norm of the gradients is one of the keys to understanding the behavior of these algorithms.

**Summary Of The Review:**

This paper focuses on bounding the expected generalization and excess risk for symmetric, deterministic algorithms for smooth losses. In particular, it focuses on full-batch gradient descent (GD).

The paper provides an alternative proof for general algorithms and then generates new proofs for full-batch (GD). These proofs depend on a quantity (path error) that depends on the norm of the loss gradients, which has been shown to be important for this algorithm generalization in the past. The bounds are sharper than previous bounds in the literature and show that GD can obtain better or equal rates than stochastic gradient descent, providing some understanding to recent experimental results.

In general, I believe the theory is interesting and the results are a valuable asset for the community. Hence, I recommend acceptance. Nonetheless, I would appreciate it if the questions raised in the weaknesses were addressed so that all the theory could be verified and if the following minor comments were taken into account.

**Minor comments and nitpicks that did not impact the score of the review**

* $R_S^\star$ is never defined, I assume it is $R_S(W_S^\star)$.

* In Theorem 7, I believe it would be clearer to say that $\eta_t \leq C/t$ where $\beta C \leq 1$, and then let $\epsilon \triangleq \beta C$.
* In Theorem 12, you forgot to write that $W_{T+1} \equiv A(S)$ and $W_{T+1}^{(i)} \equiv A(S^{(i)})$.

* In the proof of Lemma 15, it seems that in the part where $\eta_t = C/t \leq 2 / (\beta + \gamma)$, you forgot a jump of line in the inequality in the third line of the array of equations and inequalities.

* Sometimes the notation for the mathematical expressions changes. For instance, you use "The inequality (8)" in page 6 and many times in the appendix you don't use the parenthesis. Could you please homogenize the notation in this regard throughout the paper?

* The equation before inequality (37) should be $Ce^{2C\beta}T^{2C\beta} \Big(1 + \frac{1}{2C\beta} \Big) - \frac{e^{2C\beta}}{2\beta}$.

* In Appendix D: in the third line it should be "generalization error bounds for symmetric and smooth algorithms".

* In the proof of Lemma 18, it should $\sum_{j \in \mathcal{J}} f(W,z_j)$ that is convex and then the gradient of that, $h(W)$ that is $\beta |\mathcal{J}|$-Lipschitz.

* It is not immediately clear that $\epsilon_{\textnormal{opt}} \leq 3 \geta \mathbb{E}[\lVert W_1 - W_S^\star \rVert_2^2] / T$. It would be helpful to say that we obtain that from Lemma 20 where we see that $\epsilon_{\textnormal{opt}} \leq 8/3 \geta \mathbb{E}[\lVert W_1 - W_S^\star \rVert_2^2] / T$

* In the last array of expressions from appendix E, the second to last expression should be an inequality, not an equality (note that the multiplicative factor of 3 was not present in the first term of the second factor inside the square root).

* In page 29, after (105), I believe you forgot to take into account the factor 2 in that was present in $\sqrt{2 \beta \Lambda(\gamma_{loo}, T)}$. Then, in all equations following the first $\min$ is of $\frac{2\beta}{\gamma_{loo}}$ and $2T$. This carries over and I believe the final result should have the multiplicative factor $8 \sqrt{6}$ instead of $8 \sqrt{3}$.

* Sometimes the introduced extra notation harms the readability of the text instead of helping it. For instance:

  * $\Delta_T$ is used in Theorem 14, but never introduced.
  * The incorporation of $\textnormal{m}(\gamma_{loo})$ is used in only one equation to quickly move to $\textnormal{m}_{n,\gamma_{loo}}$, practically having no impact in the proof (page 30).

* Some expressions are numbered and some are not. This is okay and standard when the numbered equations are those that are referred to later on. Could you choose either to number all of them or only those that are referred to, please?

---

> ### Author Response · Authors · 2022-11-18
> **Response to Reviewer Phd7 (part 1 of 2)**
>
>  We would like to thank the Reviewer for their effort reading our paper, and for their comments, which we address as follows. The particular comments by Reviewer Phd7 were very thorough. We appreciate the effort and all the comments that significantly improved the quality of the paper. This was a very constructive and valuable review that we fully appreciate and enjoy.
>
> Q.1 ''As mentioned in the text after Theorem 3, (4) is essentially the tightest version of Theorem 2 b) of [Lei and Ying 2020b]. However, it would be nice to clarify that this is only up to constants, since the first term in (4) would be divided by 2 and the second by 4 if one were to obtain the variant in that paper.''
>
> A.1 As the reviewer suggested, we made it explicit that one can minimize (with respect to $\gamma$) the bound in Theorem 2 b) by Lei and Ying to derive the tightest version. However such choice of $\gamma$ is not a constant. In fact, one can show that $\gamma$ depends on the stability error and the expected risk of the objective at the output of the algorithm (see Theorem 2 by Lei and Ying 2020b).
>
> Q.2 ''In the remark about the particularization of Theorem 3 under the Polyak-Łojasiewicz (PL) condition, it would be nice to write what PL stands for and to cite the original references from [Polyak 1963] and [Łojasiewicz 1963] or the paper that coined the term [Karimi et al. 2016]. Also, the way the discussion after (5) is written, although correct, is a little confusing. Could you please re-write or explain further the arguments under the interpolation regime and $\beta\leq \mu n /4$?''
>
> A.2  We have included additional information to resolve this issue in the revised version of our work. We further explained the PL condition and included the suggested citation by Karimi. In fact, the PL condition corresponds to a uniformly $\mu$-PL objective %under a uniform over $\mu$-PL assumption
>   defined as $\mathbb{E} [\Vert \nabla R_S (w) \Vert^2_2] \geq 2\mu \mathbb{E} [R_S (w) - R^*_S ]$ for all $w\in\mathbb{R}^d$ (similarly to prior work by Lei and Ying, Charles and Papailiopoulos). Additionally, we explained the case of $\beta\leq \mu n /4$, by providing further discussion (see the updated version of the paper on page 5 and the footnote). If needed, we will provide additional explanation in the appendix due to space limitation.
>
> Q.3 ''After Corollary 8 it is said that (9) matches bounds in prior work, including information-theoretic bounds for the SGLD algorithm and it cites [Wang et al. 2021b]. I could not see to which bound the text refers after reading the reference, could you point that out and, in case that it needs some further modification, explain that?''
>
> A.3 Our result matches (up to a logarithmic factor) the bound in Corollary 1 by Wang et al. (see also the discussion below Corollary 1 in the prior work). In fact, for full batch SGLD, Wang et al. recover the rate $\sqrt{T}/n$.
>
> Q.4 ''An important element for the bounds of gradient methods under the Lipschitz/smooth assumptions is that they are dimension-independent when the parameter space is bounded. This is also the case here and I believe it should be mentioned clearly before showing the complexity terms that can be derived from these bounds.''
>
> A.4 We clarify that the term $\mathbb{E} [\Vert W_1 - W^*_S] \Vert^2_2 $ is $\mathcal{O}(1)$ independent of the parameters of interest (for instance $n$). We have addressed this comment below (14) on page 8 of the updated version of the paper.
>
> Q.5 ''Why does (51) hold?''
>
> A.5 The quadratic growth of $R_S (\cdot)$ is the property that we use here. If $\pi_s \in \arg \min_w R_S (w)$, then for any $w\in \mathbb{R}^d$, it is true that $ \mathbb{E} [ \Vert w - \pi_s  \Vert^2_2 ] \leq \frac{2}{\mu} \mathbb{E}[R_S (w)] - \mathbb{E}[R_S (\pi_s)]$. Since, the last holds for any $w\in\mathbb{R}$ we may choose $w=\pi_{S^{(i)}}$ to recover (51).
>
> Q.6 ''Why does (89) hold?''
>
> A.6  Observe that $\beta \geq \gamma >\gamma_{\text{loo}}$, then by replacing the number $\gamma_{\text{loo}}$ with the larger value $\beta$ we find that $$2\eta_t \frac{\beta \gamma_{\text{loo}} }{\beta+\gamma_{\text{loo}}} \geq 2\eta_t \frac{\beta \gamma_{\text{loo}}}{2\beta} =  \eta_t \gamma_{\text{loo}} .$$
>
> Q.7 In (101) you use $1-\eta_j \gamma /2$
> and then you replace that in the next array of equations for $1-\eta_j \gamma_{\text{loo}}$. Why can you do that?
>
> A.7 That was a typo and the reviewer correctly noticed it. We have corrected that. Please see (101) in the updated version of the paper.
>
> Q.8 Why is true that $\Lambda (\gamma_{\text{loo}},T  ) \leq 2T/\beta $.
>
> A.8 The function $\Lambda (\gamma_{\text{loo}},T  ) $ is decreasing with respect to $\gamma_{\text{loo}}\in (0,1)$. Thus we find the upper bound by considering $\gamma_{\text{loo}}$ and the limit of $\Lambda (\gamma_{\text{loo}},T  ) $ for $\gamma_{\text{loo}} \rightarrow 0$.

---

> > ### Author Response · Authors · 2022-11-18
> > **Response to Reviewer Phd7 (part 2 of 2)**
> >
> > Q.9 ''How do you obtain the third inequality of page 29? I am particularly surprised by the $\mathbb{E}[R_S(W^*_S)]$ before the max inside the squared root.''
> >
> >  A.9 That is correct! There is an extra term $\mathbb{E}[R_S(W^*_S)]$ that should not appear. We have corrected this part of the proof and we have updated all the generalization error bound of the strongly convex case according to this observation. The updated bounds do no involve the extra constant. We thank the review for pointing out the extra term $\mathbb{E}[R_S(W^*_S)]$.
> >
> >  Finally, we will take into consideration all the minor comments ''Minor comments and nitpicks that did not impact the score of the review'' as suggested by the reviewer (most of them have been already addressed in the updated version of the paper).

---

> ### Comment · Reviewer_Phd7 · 2022-11-21
> **Answer to response**
>
> I am happy the review was helpful. Most of my comments have been addressed. Please, find below some clarifications / further comments:
>
> **A to A1**: It is true that the choice of $\gamma$ is not constant. What we meant is that the resulting bound in *[Lei and Ying 2020b]* after optimizing over $\gamma$ is equal to (4) up to constants. That is, the constants in (4) and those in the bound resulting in optimizing Theorem 2 b over $\gamma$ are not exactly the same, even if they behave the same way. In particular, the first term in (4) would be divided by 2 and the second by 4.
>
> **A to A2**: Thanks for the update here. Nitpick: you wrote Therem 7 under (5) and in the footnote 2.
>
> **A to A3**: I see now, thanks for the clarification.

---

> > ### Author Response · Authors · 2022-12-01
> > **Resposne to additional comments and clarification (Reviewer Phd7)**
> >
> > Thank you for the additional comments and the clarification.
> >
> > The constants in (4) and the Theorem 2 by  [Lei and Ying 2020b] are different because we find an upper bound on the absolute value of the generalization error, namely $|\epsilon_{\text{gen}}|$, instead of deriving upper bounds on $\epsilon_{\text{gen}} \triangleq \mathbb{E} [  R(A(S)) -R_S (A(S))  ]$. The larger constants are introduced by (29) on page 16 (Appendix B.1, Proof of Theorem 3) In this way, we avoid any trivial upper bounds. Bounding the absolute value of the generalization error is a common approach in the literature.

---

### Official Review · Reviewer_U9d7 · 2022-10-21

**Confidence:** 3
**Clarity, Quality, Novelty And Reproducibility:** 1. I checked the proofs and statement…
**Correctness:** 4
**Technical Novelty And Significance:** 3
**Empirical Novelty And Significance:** 3
**Recommendation:** 6

**Strength And Weaknesses:**

Strength:
1. This paper proposes a generalization bound under GD regimes using stability-based techniques.
2. Although this paper generally follows the ideas of Lei and Ying (this leads to the relaxation of the Lipschitz), this paper makes some improvements.
3. The authors consider both convex and non-convex regimes in this paper.
4. The writing is clear and easy to follow.

Questions:
1. I find that the results in this paper can be deployed in non-convex cases. Could the authors re-claim the technical aspects and difficulties of this paper in non-convex cases compared to Lei and Ying?
2. The definition of path error (Def3) is for any i. However, I did not find the dependency of i in the notion $\epsilon_path$. Do the authors mean the sup over i, or expectation?

**Summary Of The Paper:**

Summary:
This paper tries to derive generalization bound using stability-based techniques under full-batch gradient descent. The results in this paper do not require any Lipschitz assumption, and the authors consider both convex and non-convex regimes. The techniques used in this paper follows Lei and Ying, where their techniques are mainly used in SGD regimes. This paper generalizes the results into GD regimes. The main difference here is Lemma~18, where the authors show a 1-expansion property for GD. Besides, the authors improve the previous analysis using Thm3.
Overall, this paper is technically sound and clean, and I would like to see it published in ICLR.


**Summary Of The Review:**

Overall, this paper is technically sound and clean, and I would like to see it published in ICLR.

---

> ### Author Response · Authors · 2022-11-18
> **Response to Reviewer U9d7**
>
> We would like to thank the Reviewer for their effort reading our paper, and for their comments, which we address as follows.
>
> Q.1 ''I find that the results in this paper can be deployed in non-convex cases. Could the authors re-claim the technical aspects and difficulties of this paper in non-convex cases compared to Lei and Ying?''
>
> A.1 Even though Lei and Ying also consider the general nonconvex case, they only show a stability bound on the estimated parameter at final iteration, but not any generalization error bounds. In fact, in their work  a generalization error bound does not appear for the general nonconvex loss case that involves a dependence on both $n$ and $T$. In contrast, we provide exact expressions for the generalization error with explicit dependence on the parameters of interest.
>
>
> Q.2 The definition of path error (Def3) is for any $i$. However, I did not find the dependency of $i$ in the notion. Do the authors mean the sup over $i$, or expectation?
>
> A.2 The definition of path error holds for any $i$ because the random variables $z_i$ are i.i.d. and the algorithm is symmetric (the output remains identical for any permutation of the input dataset). We will clarify this below the definition of path error if needed.
>
> Q.3 I cannot evaluate the technical contributions of this paper. This stops me from giving a better score. The authors may need to restate their technical contributions compared to the existing papers.
>
> A.3 We will consider a careful restatement of the technical contributions of our work. For all the regimes, nonconvex, convex, strongly convex loss, we unify techniques from prior works including the growth recursion by Hard et al. (2016) and the stability of the error $\epsilon_{\text{stab}(A)}$ by Lei and Ying (2020). To list a few technical contributions, note that the prior work by Lei and Ying 2020b) does not directly provide a proof technique for the full batch GD, as the methods are specifically constructed for the SGD with batch size one. Further, we show that one should derive the recursion with respect to  $ \Vert W_{t+1} -W_{t+1}^{(i)} \Vert_2 $ (see for instance the inequality (44)) and then consider the square of $\Vert W_{T+1} -W_{T+1}^{(i)} \Vert_2 $ (notice the difference between the big $T$ and the small $t$ --in addition to the square--). In this way, we managed to get tighter bounds in certain regimes. In contrast, for the SGD (Lei and Yin, 2020) the derivation requires to derive a recursion directly in terms of $\Vert W_{t+1} -W_{t+1}^{(i)} \Vert^2_2 $. Such an approach provides looser bounds than the full-batch technique and appears to be unavoidable for the SGD algorithm. We provide a discussion on this phenomenon after Theorem 9, on page 7, but we will consider additional explanation in the updated version of our work. Furthermore, we derive generalization error guarantees with respect to certain quantities of interest for instance the path error and optimization error, and we also show properties of these quantities as well as closed form dependence on the number of iterations. For instance, the path error involves an aggregate of the optimization error at each time $t$ as well as the interpolation error. Most of these properties and expression are uniquely associated and evaluated with respect to the full-batch GD algorithm and its optimization/stability error. Another part of our contributions (that we have already discussed in the paper) is the leave-one-out property that we showed, which applies only on the full-batch GD (see subsection 5.3 Strongly-convex objective). While we refer to the contributions of our work in Section 2 ''Related work and contributions'', we will consider additional comments in this section or in the appendix (due to space limitation), if necessary.

---

> > ### Comment · Reviewer_U9d7 · 2022-11-18
> > **A short question**
> >
> > In Q.2, could the authors clarify what the expectation (in Eqn 6) is taken over? Thanks.

---

> > > ### Author Response · Authors · 2022-11-18
> > > **Response to the short question by Reviewer U9d7**
> > >
> > > First, notice that $W_t $ is a function of $z_1 , z_2 \ldots , z_n$ for every $t$, let us use the notation $W_t \triangleq h_t (  z_1 , z_2 \ldots , z_n )$ (for some deterministic mapping $h_t (\cdot)$ because the algorithm is deterministic). Since $\Vert \nabla f(W_t ,z_{i}) \Vert$ involves the terms $h_t (  z_1 , z_2 \ldots , z_n )$ and $z_i$, $\Vert \nabla f(W_t ,z_{i}) \Vert$ is a function of $z_1 , z_2 \ldots ,z_n$. Thus the expectation is with respect to $z_1 , z_2 , \ldots ,z_n$. Of course we may also consider the expectation with respect to the set of random variables $z_1 , z_2 ,\ldots , z_n , z'_1 , z'_2 ,\ldots z'_n$, then the integration with respect to $z'_1 , z'_2 ,\ldots z'_n$ factors out because the term $\Vert \nabla f(W_t ,z_i ) \Vert $ is a function of $z_1 , z_2 \ldots ,z_n$ only, and our notation remains consistent.
> > >
> > > Given that, we proceed to further clarify the following property that holds for all $t$ and for any $j \neq i$
> > >
> > > \begin{align}
> > >  \mathbb{E} [ \Vert \nabla f(W_t ,z_{i}) \Vert^2_2 ] \equiv   \mathbb{E} [ \Vert \nabla f(h_t (  z_1 , z_2 , \ldots , z_n ) ,z_{i}) \Vert^2_2 ] &= \mathbb{E} [ \Vert \nabla f(h_t (  z_1 , z_2,  \ldots, z_i ,\ldots , z_j ,\ldots , z_n ) ,z_{i}) \Vert^2_2 ]  \\\\ &= \mathbb{E} [ \Vert \nabla f(h_t (  z_1 , z_2,  \ldots, z_j ,\ldots , z_i ,\ldots , z_n ) ,z_{j}) \Vert^2_2 ]
> > > \end{align}
> > >
> > > and the last holds because the random variables are i.i.d (relabelling). Since the function $h_t (\cdot)$ is symmetric (invariant under permutations of the input), it is true that for any $j \neq i$ and all $t$ \begin{align}
> > >  \mathbb{E} [ \Vert \nabla f(h_t (  z_1 , z_2,  \ldots, z_j ,\ldots , z_i ,\ldots , z_n ) ,z_{j}) \Vert^2_2 ] &= \mathbb{E} [ \Vert \nabla f(h_t (  z_1 , z_2,  \ldots, z_i ,\ldots , z_j ,\ldots , z_n ) ,z_{j}) \Vert^2_2 ] \\\\ &=  \mathbb{E} [ \Vert \nabla f(h_t (  z_1 , z_2 , \ldots , z_n ) ,z_{j}) \Vert^2_2 ] = \mathbb{E} [ \Vert \nabla f(W_t ,z_{j}) \Vert^2_2 ].
> > > \end{align} We combine the above to conclude that for any pair of indices $i,j$ it is true that $$ \sum_{t=1}^T \eta_t \mathbb{E} [ \Vert \nabla f(W_t ,z_{i}) \Vert^2_2 ] =  \sum_{t=1}^T  \eta_t \mathbb{E} [ \Vert \nabla f(W_t ,z_{j}) \Vert^2_2 ] = \epsilon_{\text{path}}  . $$

---

> > > > ### Comment · Reviewer_U9d7 · 2022-11-18
> > > > **Thanks for the reply!**
> > > >
> > > > Thanks for the quick reply. I get the idea now. So the expectation is taken over the training samples $z_i$.
> > > >
> > > >
> > > > There is another thing that I want to check. The most important Lemma from SGD to GD, in my opinion, is Lemma 18. I want to check whether this is first derived in this paper, or is an existing result (I did not find the reference in the Appendix). Thanks!
> > > >
> > > > Besides, the authors mentioned that ``Most of these properties and expressions are uniquely associated and evaluated with respect to the full-batch GD algorithm and its optimization/stability error.'' This is a good point to state technical contributions. However, I am still not convinced that the techniques used in this paper are indeed **uniquely** applied to GD algorithms. For example, it seems that one can also use a similar proof line (bounding path-error) in SGD regimes. The authors could correct me if I am wrong.

---

> > > > > ### Author Response · Authors · 2022-11-30
> > > > > **Response to additional comments and questions (Reviewer U9d7)**
> > > > >
> > > > > Lemma 18 does not exist in prior works and we thank the reviewer for pointing out this as a part of the contributions.
> > > > >
> > > > > To provide a complete picture of the importance of Lemma 18 and Lemma 21, and how those impact the derivation of the corresponding generalization error bounds, let us first discuss the overall technical approach. We derive generalization error guarantees for general non-Lipschitz nonconvex losses. This improves upon results in prior works since the work by Hardt et al. relies on the Lipschitz loss assumption, while Lei and Ying do not derive explicit generalization error bounds for general smooth nonconvex losses (in the case of SGD). To show generalization error guarantees for full-batch GD, we consider the solution of the growth recursion in (44). Such an approach is not applicable to the SGD because of the random selection rule at each iteration (see C.7 page 17 Lei and Ying 2020b, ArXiv version). Specifically, we solve the recursion with respect to the $\ell_2$ norm and then evaluate the square (of the final iteration stability error). This approach leads to a term of the  order of $1/n^2$ in our stability bounds, in contrast to the order-$1/n$ term that appears in Theorem 3, (4.4) in Lei and Ying 2020b. In this way, we exploit the deterministic nature of the algorithm to derive tighter generalization error bounds (for instance by an order of magnitude in the convex case, see the discussion bellow Theorem 9 in our paper).
> > > > >
> > > > > Additionally, for convex loss and strongly convex objective, two novel parts of our analysis that are also dedicated to the full-batch GD algorithm are Lemmas 18 and 21 in Appendices E and F. As Reviewer U9d7 observed, such bounds do not appear in prior works (although an elementary argument regarding non-expansiveness of the SGD based on co-coersivity appears in Hard et al., ArXiv version, page 31). In fact, both lemmas 18 and 21 involve the gradient of the leave-one-out empirical loss $R_{S^{-i}}(w)\triangleq \sum^n_{j=1 , j\neq i} f(w;z_j)/n$, that uniquely appears in the analysis of full-batch GD. Lemmas 18 and 21 are necessary for the derivation of the bounds in the inequality (81) on page 24 and the inequality (102) on page 27, respectively. These two parts of the analysis enable us to combine the recursion with respect to the $\ell_2$-norm (as discussed in the previous paragraph) with Lemmas 18 and 21, for the convex and strongly convex case, respectively. Finally, through Lemma 21 we recover the stability guarantees of the convex case when $\gamma_{\text{loo}}\rightarrow 0$, while in the prior work by Lei and Ying 2020b this is not the case.

---

> > > > > > ### Comment · Reviewer_U9d7 · 2022-12-07
> > > > > > **Thanks for the reply**
> > > > > >
> > > > > > Thanks for the detailed reply. I remain positive on this paper, given that Lemma 18 is new.

---

> > > > ### Comment · Reviewer_U9d7 · 2022-11-28
> > > > **Hi authors?**
> > > >
> > > > Hi, could the authors check the response "thanks for the reply"? This may help me better evaluate the submission. Thanks.

---

> > > > > ### Author Response · Authors · 2022-12-01
> > > > > **Thank you for the reminder!**
> > > > >
> > > > > Thank you for the reminder. We have replied to the initial comment. Please see our response below ''Response to additional comments and questions (Reviewer U9d7)''

---

### Official Review · Reviewer_zLoh · 2022-10-25

**Confidence:** 3
**Clarity, Quality, Novelty And Reproducibility:** The presentation is clear and easy to…
**Correctness:** 3
**Technical Novelty And Significance:** 3
**Empirical Novelty And Significance:** Not applicable
**Recommendation:** 5

**Strength And Weaknesses:**

Strength:
The paper provides tighter generalization bounds for Gradient Descent via stability argument, which can serve as basic results for this type of references.
The paper is clearly written, and references are covered sufficiently.

Weakness:
There are no convincing insights for such bounds. Though the generalization bounds for GD are comparable to that of SGD with much fewer iterations, GD cost much more computation than SGD for each iteration.  Even for the literature where GD achieves the "same" excess risk as SGD, GD requires much more pass of data. There is no sufficient motivation to advocate the use of GD over SGD.



The expectations in many definitions and theorems all have the same meaning. Please do specify what is expected over.

**Summary Of The Paper:**

The paper provides sharp generalization bounds for the full-batch Gradient Descent algorithm on smooth losses. It builds upon the stability argument and risk decomposition. It derives a generalization error specifically for nonconvex, convex and strongly convex cases with smoothness assumption.

**Summary Of The Review:**

The paper provides improved generalization analysis for gradient descent for various cases under smoothness assumption. However, the improvement on the number of iterations for GD over SGD is not surprising, which mainly attributes to the less noise in GD and the better convergence rate GD. The paper does not provide insightful implication for practice.

---

> ### Author Response · Authors · 2022-11-18
> **Response to Reviewer zLoh**
>
> We would like to thank the Reviewer for their effort reading our paper, and for their comments, which we address as follows.
>
>   Q.1  ''There are no convincing insights for such bounds. Though the generalization bounds for GD are comparable to that of SGD with much fewer iterations, GD cost much more computation than SGD for each iteration. Even for the literature where GD achieves the "same" excess risk as SGD, GD requires much more pass of data. There is no sufficient motivation to advocate the use of GD over SGD.''
>
>   A.1 (This point was also raised by Reviewer A39n; please see above) We have already addressed this question on page 3, please see also the footnote. For completeness, we repeat that the GD indeed requires more computation than SGD, and SGD is the best option in most practical scenarios. However, the directional step of GD can be evaluated in parallel. As a consequence, GD would be more iteration-efficient than SGD (in terms of running time) if some parallel computation is available, but at the expense of extra computational cost. In fact, we highlight the trade-off between performance and resource allocation on page 3 of the paper. We will add a remark to explicitly clarify the above trade-off in the revised version of the paper.
>
> Additionally, note that there is a long debate on the literature on whether inherent randomization of training algorithms (such as SGD) is necessary or fundamental for achieving good generalization (see for instance Hard et al., 2016, Charles and Papailiopoulos, 2018, and several other more recent studies). We show in our work that randomization is not necessary, and in certain cases full-batch GD achieves tighter upper bounds on excess risk (of course, at the expense of additional computation). We would like to kindly point out that this theoretical question (i.e., whether randomization is needed or not) was initially our motivation for analyzing the generalization performance of full-batch GD. Our bounds improve the state of the art in this direction.
>
>   In particular to the comment made by the reviewer, one part of the contribution of the paper is to show that the generalization error is not hindered by lack of algorithmic randomization.
>   In practice, our results should motivate the use of mini-batch SGD with potentially larger batches, which is "in the middle" between fully randomized SGD (which is by the way usually the version of SGD considered in most theory papers studying algorithmic generalization) and full-batch GD. A more detailed study of this can be the subject for future research.
>
>   Q.2 ''The expectations in many definitions and theorems all have the same meaning. Please do specify what is expected over.''
>
>   A.2 The expectations are over the data-set which is the only random element in this work, since the algorithm is deterministic. We had provided already a comment regarding the expectation in Section 4, ''Symmetric algorithm and smooth loss''. We had highlighted that ''expectation is over the random variables $z_1,\ldots,z_n , z'_1,\ldots ,z'_n$ in the main part of Section 4. We will make this more explicit in the revised version of the paper.
>
>   Q.3 ''The paper provides improved generalization analysis for gradient descent for various cases under smoothness assumption. However, the improvement on the number of iterations for GD over SGD is not surprising, which mainly attributes to the less noise in GD and the better convergence rate GD. The paper does not provide insightful implication for practice.''
>
>   A.3 Of course, in terms of optimization, GD can converge faster than SGS and this is indeed not surprising. However, it is not clear at all that an algorithm that converges faster can generalize better. Indeed, prior works claim that full-batch GD does not generalize and that the randomization of the algorithm is necessary and fundamental for generalization (Hardt et al., 2016; Charles \& Papailiopoulos, 2018). We would like to note that most of the theory  on the generalization power of SGD including Hard et al. 2016 and Lei and Yin 2020 consider only the case of batch size equal to one. One can claim that this is not interesting in practice because SGD with single example batch sizes may not work. We believe that prior works and our work provide a path to explore and understand practical situations and close the gap.

---

### Official Review · Reviewer_A39n · 2022-11-04

**Confidence:** 4
**Correctness:** 4
**Technical Novelty And Significance:** 3
**Empirical Novelty And Significance:** Not applicable
**Recommendation:** 5

**Clarity, Quality, Novelty And Reproducibility:**

The problem formation and the results are stated clearly and the summary tables provide a good and clear summary of contributions.
In terms of originality and novelty, I believe the techniques are close to the Lei & Ying, 2020b paper.

**Strength And Weaknesses:**

I believe the main interesting result is regarding the strongly convex case where the number of iterations decreases from $n$ to $\log(n)$. While this is expected (because of the GD convergence rate for the ERM problem), it shows the benefit of full-batch GD in this case.

Regarding the convex case, for the case that the interpolation error is zero, GD and SGD have similar performance, while for the case that the interpolation error is non-zero, full-batch GD leads to $\sqrt{n}$ improvement. However, one should note that full-batch GD requires $n$ times more gradient computation, and so overall complexity is worse than SGD. I would appreciate the authors' comments on this matter.

Regarding the non-convex case, I am slightly confused by the assumption on learning rate ($\eta_t \leq \Omega(1/t)$). I believe the convergence rate to a stationary point would be very slow in this case. I would appreciate it if the authors comment on the convergence rate under this assumption.

**Summary Of The Paper:**

This paper studies the generalization of full-batch gradient descent for strongly convex, convex, and nonconvex smooth but possibly non-Lipschitz functions. In particular, the authors extend the existing results for SGD to full-batch GD in all cases and compare the excess risk bounds. More importantly, the authors derive these results without the Lipschitz assumption, which is customary in the stability analysis.

**Summary Of The Review:**

Overall, I find this paper interesting as it drops the Lipschitz assumption for the generalization analysis of full-batch GD. However, I have two main questions (mentioned above) regarding the convex case's overall complexity and the nonconvex's convergence rate. I would appreciate it if the authors could clarify these two matters (and I am willing to increase my score given the authors' response).

---

> ### Author Response · Authors · 2022-11-18
> **Response to Reviewer A39n**
>
> We would like to thank the Reviewer for their effort reading our paper, and for their comments, which we address as follows.
>
> Q.1 ''Regarding the convex case, for the case that the interpolation error is zero, GD and SGD have similar performance, while for the case that the interpolation error is non-zero, full-batch GD leads to $\sqrt{n}$ improvement. However, one should note that full-batch GD requires times more gradient computation, and so overall complexity is worse than SGD. I would appreciate the authors' comments on this matter.''
>
> A.1 We have already addressed this question on page 3, please see also the footnote. For completeness, we repeat that the GD indeed requires more computation than SGD, and SGD is the best option in most practical scenarios. However, the directional step of GD can be evaluated in parallel. As a consequence, GD would be more iteration-efficient than SGD (in terms of running time), if some parallel computation is available, but at the expense of extra computational cost. In fact, we highlight the trade-off between performance and resource allocation on page 3 of the paper. We will add a remark to explicitly clarify the above trade-off in the revised version of the paper.
>
> Additionally, note that there is a long debate on the literature on whether the inherent randomization of the training algorithms (such as SGD) is necessary or fundamental for achieving good generalization (see for instance Hard et al., 2016, Charles and Papailiopoulos, 2018, and several other more recent studies). We show in our work that randomization may not be necessary, and in certain cases full-batch GD achieves tighter upper bounds on excess risk (of course, at the expense of additional computation). We would like to kindly point out that this theoretical question (i.e., whether randomization is needed or not) was initially our motivation for analyzing the generalization performance of full-batch GD. Our bounds improve the state of the art in this direction.
>
>
> Q.2 Regarding the non-convex case, I am slightly confused by the assumption on learning rate ($\eta_t = \Omega ( 1/t)$). I believe the convergence rate to a stationary point would be very slow in this case. I would appreciate it if the authors comment on the convergence rate under this assumption.
>
> A.2 We believe that this is a reasonable concern, as convergence rates for diminishing step-sizes are unknown for the general nonconvex case (to the best of our knowledge). That said, the step-size choice of $\eta_t =1/t$ is a standard choice in algorithmic generalization; in particular, see also the analysis for the nonconvex case in the seminal prior work by Hardt et al. (2016). Of course, while a choice of a  "large" but fixed step size $\eta_t =1/\beta$ would give faster convergence to a stationary point (a case for which rates of convergence have been studied), the algorithm appears to be unstable --in terms of uniform algorithmic stability-- (note that this is also holds SGD) and fails to provide efficient generalization bounds in terms of the number of iterations $T$. In practice though, we may heuristically choose a small fixed step size $\eta$ that is much smaller than the "larger" choice $1/\beta$.

---

### Author Response · Authors · 2022-11-18
**Generic Response**

We would like to thank the reviewers for their detailed comments on our paper. We believe that the feedback and the subsequent revision will greatly improve the manuscript. All typos and minor comments will be addressed accordingly. Within the nine page limit we have tried to address most of the comments by the reviewers. If the reviewers suggest additional changes, we are planning to address all comments as best as possible, either in the main body of the paper, or in the appendix, in a revised version after the reviewing process. We continue by providing our responses to each reviewer.

---

### Author Response · Authors · 2022-11-27
**Feedback to our initial responses**

While we are preparing our answers to further questions raised by the reviewers, we would like to let all know that we are waiting and would kindly appreciate feedback to our responses to the initial comments to our work, so that we can address all concerns adequately and in time. We have already posted our initial replies and we will proceed by addressing any additional comments or questions.

---

### Decision · Program_Chairs · 2023-01-20

**Decision:**

Accept: poster

**Justification For Why Not Higher Score:**

Reviews are highly mixed

**Justification For Why Not Lower Score:**

Overall the most active reviewers were satisfied and come out in support of the paper, and therefore I find it natural to accept.

**Metareview: Summary, Strengths And Weaknesses:**

This paper provides a wide array of stability-based generalization analyses of gradient descent.  While the reviews were somewhat mixed, overall the internal discussions focused on a variety of new contributions within the proofs, and therefore I am happy to recommend acceptance for this paper.

**Note From Pc:**

if the above contains the word "oral" or "spotlight" please see: "oral" presentation means -> notable-top-5% and "spotlight" means -> notable-top-25%. As stated in our emails, we are disassociating presentation type from AC recommendations